# Revealing exciton masses and dielectric properties of monolayer semiconductors with high magnetic fields

M. Goryca[1], J. Li [ID] [1], A.V. Stier[1], T. Taniguchi[2], K. Watanabe [ID] [2], E. Courtade[3], S. Shree[3], C. Robert[3], B. Urbaszek [ID] [3], X. Marie[3] & S.A. Crooker [ID] [1]

In semiconductor physics, many essential optoelectronic material parameters can be experimentally revealed via optical spectroscopy in sufficiently large magnetic fields. For monolayer transition-metal dichalcogenide semiconductors, this field scale is substantial—tens of teslas or more—due to heavy carrier masses and huge exciton binding energies. Here we report absorption spectroscopy of monolayer $MoS_2$, $MoSe_2$, $MoTe_2$, and $WS_2$ in very high magnetic fields to 91 T. We follow the diamagnetic shifts and valley Zeeman splittings of not only the exciton's $1s$ ground state but also its excited $2s, 3s, \ldots, ns$ Rydberg states. This provides a direct experimental measure of the effective (reduced) exciton masses and dielectric properties. Exciton binding energies, exciton radii, and free-particle bandgaps are also determined. The measured exciton masses are heavier than theoretically predicted, especially for Mo-based monolayers. These results provide essential and quantitative parameters for the rational design of opto-electronic van der Waals heterostructures incorporating 2D semiconductors.

[1] National High Magnetic Field Laboratory, Los Alamos National Lab, Los Alamos, NM 87545, USA. [2] National Institute for Materials Science, Tsukuba, Ibaraki 305-0044, Japan. [3] Universite de Toulouse, INSA-CNRS-UPS, LPCNO, 135 Av. Rangueil, 31077 Toulouse, France. Correspondence and requests for materials should be addressed to S.A.C. (email: crooker@lanl.gov)

In their bulk form, the transition-metal dichalcogenide (TMD) materials $MoS_2$, $MoSe_2$, $MoTe_2$, $WS_2$, and $WSe_2$ are indirect-gap semiconductors[1,2]; however they become direct-gap semiconductors when thinned down to a single monolayer[3,4]. This phenomenon, together with the remarkable valley-specific optical selection rules that emerge in the atomically thin limit[5], has galvanized tremendous interest in the underlying properties of this new 2D semiconductor family[6–8]. With a view towards future generations of ultrathin devices based on van der Waals assembly[9] of TMD monolayers, a quantitative understanding of their intrinsic material parameters is of obvious and critical importance. For applications in optoelectronics, properties related to the fundamental electron–hole quasiparticle excitations by light—that is, excitons—are particularly relevant: the exciton's mass, size, binding energy, oscillator strength, and lifetime are key variables, as are the dielectric screening properties and the free-particle bandgap of the monolayer itself. These material parameters constitute necessary inputs for any rational design and engineering of functional van der Waals heterostructures.

To date, however, many of these fundamental parameters are still assumed from density functional theory calculations[10–12] and have not been experimentally measured. This is particularly true for the exciton masses and the effective dielectric screening lengths in the monolayer TMD semiconductors—essential ingredients for realistic optoelectronic device models. In principle, however, these and other crucial material properties can be directly accessed via optical spectroscopy of excitons in large magnetic fields. As myriad studies over the past several decades have amply demonstrated in III–V, in II–VI, and in various layered semiconductors[13–17], the diamagnetic shifts of the exciton transitions in sufficiently high magnetic fields can directly reveal the exciton's mass, independent of any model. Moreover, quantitative analysis becomes especially straightforward and increasingly accurate when the less-strongly bound excited states of the exciton (i.e., the optically allowed 2s, 3s, ..., ns Rydberg states) are also observable, in which case the relevant dielectric properties of the material itself can also be determined.

Until quite recently, the optical quality of TMD monolayers was typically not sufficient to achieve very narrow exciton absorption lines or well-resolved Rydberg exciton states. However, by encapsulating TMD monolayers between atomically smooth hexagonal boron nitride (hBN) slabs, several groups have shown that the optical quality of TMD monolayers can be dramatically improved to the point where exciton spectral lines approach the narrow homogeneously broadened limit (1–2 meV) and where excited Rydberg exciton states become clearly visible[18–26]. Consequently, the first magneto-optical studies of Rydberg exciton diamagnetic shifts were performed only recently on high-quality hBN-encapsulated $WSe_2$ monolayers[27]: Using pulsed magnetic fields up to 65 T, the diamagnetic shifts of the ground state (1s) and the excited 2s, 3s, and 4s states of the neutral exciton were measured and used to determine the exciton's mass, size, and binding energy.

However, the exciton masses and dielectric properties have not been experimentally determined for any of the other members of the monolayer TMD semiconductor family, and general trends have not been established. This represents an especially challenging task for the molybdenum-based monolayers, for which electron, hole, and exciton masses are theoretically predicted to be even heavier than those of their tungsten-based counterparts[10–12], potentially requiring even larger magnetic fields. An experimental determination of these basic material parameters is particularly urgent in view of very recent electrical transport measurements in n-type $MoS_2$ and $MoSe_2$ monolayers that reveal an anomalous and surprisingly large electron mass that exceeds theoretical predictions by nearly a factor of two[28,29]. Whether these unexpectedly heavy electron masses result from interactions in the high-density electron gas, or are instead an unanticipated but intrinsic material property, remains an open and crucial question.

To address these gaps in our current knowledge, here we present detailed magneto-absorption studies of $MoS_2$, $MoSe_2$, $MoTe_2$, and $WS_2$ monolayers in extremely high magnetic fields up to 91 T and at low temperatures (4 K). Magneto-optical spectroscopy provides an important and complementary approach to transport techniques, by allowing to study material parameters in the limit of zero doping. The use of hBN-encapsulated monolayers allows to observe and follow the diamagnetic shifts (and also the valley Zeeman splittings) of not only the 1s ground state of the neutral excitons but also their excited ns Rydberg states. The energies and diamagnetic shifts of the ns excitons provide a first experimental measure of the exciton's mass and also the dielectric screening length in these monolayer semiconductors. Remarkably, the exciton masses are heavier than theoretically predicted, particularly for the entire family of Mo-based monolayers. Moreover, from the data we also determine other important optoelectronic properties including exciton binding energies, exciton radii, and free-particle bandgaps. These results, summarized in Table 1, provide essential and quantitative material parameters that are necessary for the rational design of van der Waals heterostructures incorporating 2D semiconductor monolayers.

## Results

**Samples**. To prepare high optical quality monolayer samples for magneto-absorption studies in pulsed magnetic fields, exfoliated TMD monolayers are sandwiched between slabs of exfoliated hexagonal boron nitride (hBN) using a dry-stamp method[19]. The thicknesses of the hBN slabs are selected to maximize the absorption of light by the exciton resonances in the TMD monolayer[24]. As shown in Fig. 1a, each van der Waals structure is assembled directly over the 3–4 $\mu$m diameter core of a single-mode optical fiber. Crucially, this ensures a rigid drift- and vibration-free alignment of the optical path through the TMD monolayer during the experiment—conditions which can otherwise be experimentally challenging in high-field optical studies of

---

**Table 1 Fundamental optoelectronic material parameters of monolayer TMD semiconductors**

| Material | $m_r$ ($m_0$) | $E_b$ (meV) | $E_{gap}$ (eV) | $\kappa$ | $r_0$ (nm) | $r_{1s}$ (nm) |
|---|---|---|---|---|---|---|
| hBN | $MoS_2$ | hBN | $0.275 \pm 0.015$ | 221 | 2.160 | 4.45 | 3.4 | 1.2 |
| hBN | $MoSe_2$ | hBN | $0.350 \pm 0.015$ | 231 | 1.874 | 4.4 | 3.9 | 1.1 |
| hBN | $MoTe_2$ | hBN | $0.360 \pm 0.040$ | 177 | 1.352 | 4.4[a] | 6.4 | 1.3 |
| hBN | $WS_2$ | hBN | $0.175 \pm 0.007$ | 180 | 2.238 | 4.35 | 3.4 | 1.8 |
| hBN | $WSe_2$ | hBN[b] | $0.20 \pm 0.01$ | 167 | 1.890 | 4.5 | 4.5 | 1.7 |

Experimentally determined values of the exciton reduced mass $m_r$, the 1s exciton binding energy $E_b$, the free-particle bandgap $E_{gap}$, the dielectric screening parameters $r_0$ and $\kappa$, and the root-mean-square radius of the 1s exciton $r_{1s}$. Typical error bars on experimental values of $E_b$ and $E_{gap}$ are ±3 meV. Typical error bars on values of $r_0$ and $r_{1s}$ are ±0.1 nm, except for MoTe$_2$, where they are ±0.3 nm
[a]The value of $\kappa$ for MoTe$_2$ is assumed to be 4.4 and is not a fitting parameter (see text for details)
[b]Values for hBN-encapsulated WSe$_2$ are taken from ref. [27]

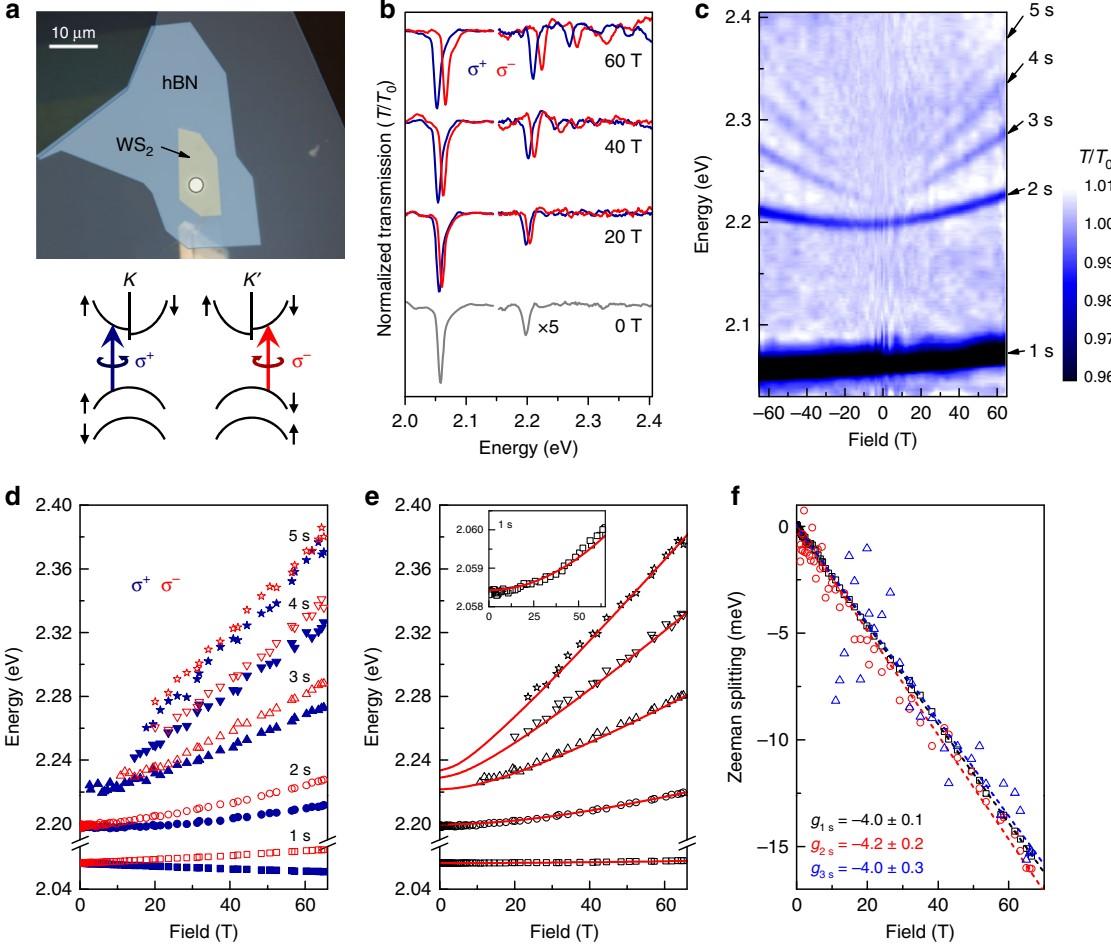

**Fig. 1** Magneto-optical spectroscopy of monolayer $WS_2$. **a** Image of a sample/fiber assembly: An exfoliated TMD monolayer, sandwiched between hBN slabs, is constructed over the $3-4$ $\mu$m diameter core of a single-mode optical fiber (white circle). The assembly is mounted in helium exchange gas at 4 K in the bore of a pulsed magnet. The diagram depicts the optical selection rules in the $K$ and $K'$ valleys. **b** Normalized transmission spectra ($T/T_0$) of monolayer $WS_2$ at selected magnetic fields $B=0$, 20, 40, and 60 T. Blue/red curves indicate $\sigma^+/\sigma^-$ circular polarization (optical transitions in the $K/K'$ valleys, respectively). **c** Intensity map of all the transmission spectra, from $-65$ to $+65$ T. Excellent sample quality allows observation of the 2s, 3s, 4s, and 5s excited Rydberg states of the neutral A exciton. **d** Energies of the $1s-5s$ excitons for both $\sigma^{\pm}$ polarizations. **e** Average energy of the $\sigma^{\pm}$ transitions for each $ns$ state, $\frac{1}{2}(E_{\sigma+} + E_{\sigma-})$, reveals distinct diamagnetic shifts. Solid lines show calculated energies using the Rytova–Keldysh model described in the text. Parameters: $m_r = 0.175\ m_0$, $r_0 = 3.4$ nm, $\kappa = 4.35$, and $E_{gap} = 2.238$ eV. Inset: Expanded plot of the 1s exciton energy, showing its small quadratic diamagnetic shift. **f** The Zeeman splitting of the 1s, 2s, and 3s exciton states ($E_{\sigma+} - E_{\sigma-}$); dashed lines depict linear fits

small samples. A total of four different $MoS_2$, two $MoSe_2$, two $MoTe_2$, and two $WS_2$ monolayer structures were studied.

**Monolayer $WS_2$.** We first present and discuss the magneto-optical spectra from the $WS_2$ monolayers, where the exciton absorption signals are very strong and the field-induced diamagnetic shifts are large. Moreover, owing to the very large ($\approx 400$ meV) spin–orbit splitting of the valence bands at the $K/K'$ points in tungsten-based TMD monolayers, the excited $ns$ Rydberg states of the neutral A exciton always remain well below the absorption band of the higher energy B exciton, which simplifies the data analysis (as shown later, this is not the case for the molybdenum-based TMD monolayers). For completeness, in this section we also describe the data analysis and modeling procedure.

Signatures of excited Rydberg excitons in monolayer $WS_2$ were first revealed by optical reflection spectroscopy at zero magnetic field by Chernikov et al.[30]. Subsequently, the diamagnetic shift and valley Zeeman splitting of the A exciton ground state (A:1s) were measured via polarized magneto-reflection to 65 T by Stier et al.[31]. More recently, charged excitons and also the excited A:2s exciton in monolayer $WS_2$ were studied up to ~30 T by

photoluminescence[32,33]. However, in all these field-dependent studies the sample quality was not sufficient to observe highly excited Rydberg states, and therefore the exciton's effective (reduced) mass, $m_r = m_e m_h/(m_e + m_h)$, could not be experimentally determined. Rather, it was always assumed that $m_r \approx 0.15 - 0.16\ m_0$ based on leading density functional theories[10,11]. Together with the measured diamagnetic shifts, this allowed estimates of the exciton's size and binding energy[31–33].

Here we observe the field-induced shifts of the 1s, 2s, 3s, 4s, and 5s states of the neutral A exciton up to 65 T. Crucially, these data provide a quantitative measurement of $m_r$ in monolayer $WS_2$. Equally importantly, these results also allow a detailed and quantitative comparison with the popular Rytova–Keldysh model that describes the non-hydrogenic electrostatic potential $V(r)$ between an electron and hole in a 2D material, from which the dielectric screening properties, exciton binding energy and size, and free-particle bandgap can also be determined.

Figure 1b shows normalized transmission spectra, $T/T_0$, at selected magnetic fields $B$ for both $\sigma^{\pm}$ circular polarizations (where $T_0$ is a background reference spectrum). At $B=0$ two narrow (width $\approx 10$ meV) absorption lines are visible at 2.058 and

2.199 eV, and with increasing field additional higher energy and rapidly shifting features are observed. To most easily visualize these trends, Fig. 1c shows an intensity map of all the $T/T_0$ spectra from $-65$ T to $+65$ T. A systematic series of absorption features are clearly resolved, that correspond (as confirmed below) to the 1s ground state and the excited ns Rydberg states of the neutral A exciton. We emphasize that the highly excited Rydberg excitons (4s, 5s) can only be clearly resolved in large $B$. Primarily this is because large magnetic fields increase the exciton's absorption oscillator strength by providing an additional effective confining potential which 'squeezes' the exciton wavefunction, thereby increasing the electron–hole overlap particularly for highly excited states that are only weakly bound at $B=0$ (ref. [13]). This highlights a further benefit of very large magnetic fields for optical spectroscopy of Rydberg excitons in semiconductors. We also note that the absence of any absorption features related to charged excitons confirms close-to-zero doping levels in the monolayer. Finally, the neutral B exciton in monolayer $WS_2$, not shown, appears as a broader absorption feature at higher energies near 2.45 eV and is not discussed further.

Figure 1d shows the field-dependent energies of all the absorption features. All exciton states exhibit a comparable Zeeman splitting between the $\sigma^+$ and $\sigma^-$ polarized optical transitions in the $K$ and $K'$ valleys, respectively, where the splitting $E_{\sigma+} - E_{\sigma-} = g\mu_B B$. The exciton g-factors are all close to $g_{ns} \approx -4$ (see Fig. 1f), in line with earlier studies of the 1s exciton in $WS_2$ monolayers[31,33,34] and in accordance with the similar Zeeman splittings of all the ns Rydberg excitons that was recently reported in $WSe_2$ monolayers[27,35].

Most importantly, Fig. 1e shows the average transition energy of each exciton state, $\frac{1}{2}(E_{\sigma+} + E_{\sigma-})$, which reveals the unique diamagnetic blueshift of each state. While the most tightly bound exciton states (1s, 2s) shift quadratically with $B$, the key point is that the most weakly bound exciton states (4s, 5s) shift much more linearly in high fields. As discussed below, in this regime the exciton's reduced mass $m_r$ can be inferred directly from the shifts and energy separations of these states, independent of any model or other material parameters such as the dielectric properties of the monolayer and its surrounding environment.

**Excitons in weak- and strong-field limits.** For completeness, we briefly reiterate how magnetic fields influence exciton energies, a topic that is very well established from fundamental semiconductor physics[13,14,27,36–38]: At small $B$ where the characteristic magnetic (cyclotron) energy $\hbar\omega_c = \hbar eB/m_r$ is much less than the Coulomb (exciton binding) energy, the lowest-order correction to the ns exciton's energy is the quadratic diamagnetic shift, $\Delta E_{ns}^{dia} = e^2\langle r_\perp^2\rangle B^2/8m_r = \sigma_{ns}B^2$. Here, $\sigma_{ns}$ is the diamagnetic coefficient, $r_\perp$ is a radial coordinate in the plane normal to $B$, and $\langle r_\perp^2\rangle = \langle\psi_{ns}(r)|r_\perp^2|\psi_{ns}(r)\rangle$ is an expectation value computed over the envelope wavefunction of the ns exciton. The root-mean-square (rms) radius of the ns exciton is therefore $r_{ns} = \sqrt{\langle r_\perp^2\rangle} = \sqrt{8m_r\sigma_{ns}}/e$. Thus, if the reduced mass $m_r$ is known (or assumed from theory), then the measured quadratic shift depends only on the exciton's size—which, in turn, typically depends strongly on the dielectric properties of the material and its immediate surroundings[39–41]. The inset of Fig. 1e shows that the shift of the most tightly bound 1s exciton in monolayer $WS_2$ does indeed increase quadratically over the entire 65 T field range. This is expected, because the 1s exciton in this structure has a large Coulomb binding energy (>150 meV, as shown below) and remains firmly in this weak-field limit even at 65 T (for comparison, the characteristic cyclotron energy $\hbar\omega_c \approx 40$ meV at 65 T, if $m_r = 0.2\ m_0$).

Conversely, in the opposite limit where $\hbar\omega_c$ greatly exceeds the Coulomb energy (i.e., for strong $B$ fields and/or weakly bound excitons), the exciton binding energy is negligible compared to the separation between the electron or hole Landau levels (LLs). To a good approximation, optically allowed interband transitions therefore occur between the linearly dispersing free-particle LLs in the valence and conduction bands, and the transition energies of the ns exciton states increase roughly linearly with $B$ as $(N + \frac{1}{2})\hbar\omega_c$, ignoring spin effects. Following convention, $N$ ($= 0$, 1, 2, ...) labels the transition number starting from the lowest (ground) state, and therefore $n \equiv N + 1$. Note that this behavior holds not only for conventional semiconductors but also for monolayer TMD semiconductors which obey the massive Dirac Hamiltonian and therefore have 0th free-particle LLs that are pinned to the bottom of the conduction band and top of the valence band in the $K$ and $K'$ valleys, respectively[42,43] (for additional details see, for example, ref. [44] or the Supporting Information in ref. [27]). In other words, the average transition energy of the ns exciton state, $\frac{1}{2}(E_{\sigma+} + E_{\sigma-})$, or equivalently the average of the two corresponding transitions in the $K$ and $K'$ valley, increases with a slope that approaches $(N + \frac{1}{2})\frac{\hbar\omega_c}{B} = (N + \frac{1}{2})\frac{\hbar e}{m_r}$ in the high-field limit (where, as above, $n \equiv N + 1$).

As discussed previously[27], the slope and the separation of the most weakly bound excitons therefore provide unambiguous and increasingly stringent upper and lower bounds on $m_r$, respectively, in the limit of large $B$. Crucially, this is independent of any other material parameter or model of the electrostatic potential. For monolayer $WS_2$, the data in Fig. 1e reveal a high-field slope of the 5s exciton of 2.65 meV T$^{-1}$, which should asymptotically approach (from below) a value of $\frac{9}{2}\frac{\hbar e}{m_r}$ in the high-field limit. Conversely, the energy separation between the 4s and 5s states, which equals 47 meV at 65 T, should asymptotically approach (from above) a value of $\hbar\omega_c = \hbar eB/m_r$ in the limit of very high fields. From these experimental values alone we can conclude that $m_r$ lies approximately midway between the lower and upper bounds of 0.160 $m_0$ and 0.197 $m_0$.

**Modeling exciton energies with the Rytova–Keldysh potential.** More refined estimates of $m_r$ are achieved by modeling the field dependence of all the ns exciton energies. To this end we numerically solve the eigenvalue (Schrödinger) equation $H\psi_{ns}(r) = E_{ns}\psi_{ns}(r)$ to find the energies and wavefunctions of the ns exciton states in an applied magnetic field. For radially symmetric s-type exciton states in 2D in perpendicular fields, $H = -(\hbar^2/2m_r)\nabla_r^2 + e^2B^2r^2/8m_r + V(r)$, where $r$ is the relative distance between the electron and hole and $V(r)$ is the electrostatic potential between the electron and hole. Very general formulations of $V(r)$ can be derived for 2D materials based on first principles[39,45,46]. Here we adopt the popular Rytova–Keldysh potential[47–49] that has been shown to accurately describe this potential in a thin semiconductor slab that is confined between dielectrics with $\varepsilon_{top}$ and $\varepsilon_{bottom}$, and provide good agreement with measured ns exciton energies[27,30]. In the limit of an infinitely thin 2D slab and reasonably large dielectric contrast between the 2D material and the surrounding materials, the Rytova–Keldysh potential can be expressed analytically as

$$V_{RK}(r) = -\frac{e^2}{8\varepsilon_0 r_0}\left[H_0\left(\frac{\kappa r}{r_0}\right) - Y_0\left(\frac{\kappa r}{r_0}\right)\right], \qquad (1)$$

where $H_0$ and $Y_0$ are the Struve function and Bessel function of the second kind. The average dielectric constant of the encapsulating materials is given by $\kappa = \frac{1}{2}(\varepsilon_{top} + \varepsilon_{bottom})$, and the dielectric properties of the 2D layer itself are characterized by its

screening length $r_0 = 2\pi\chi_{2D}$, where $\chi_{2D}$ is the material's 2D polarizability. At large electron–hole separations $r \gg r_0$, $V_{RK}(r)$ scales as $-1/\kappa r$, similar to a bulk semiconductor. However when $r \leq r_0$, $V_{RK}(r)$ begins to diverge much more slowly and tends towards $\log(r/r_0)$. This leads to a spectrum of $ns$ exciton states with energy separations that deviate markedly from a conventional hydrogen-like Rydberg spectrum, especially for the most tightly bound $1s$ and $2s$ excitons.

**Material parameters for monolayer WS$_2$.** Using $V_{RK}$ we model the exciton spectrum shown in Fig. 1e, where the solid red lines show the calculated energies. As noted above, at large $B$ the most weakly bound excitons ($4s$, $5s$) are approaching the strong-field limit, where their shifts are determined primarily by $m_r$ and are virtually independent of the dielectric parameters $\kappa$ and $r_0$. From the fits we determine $m_r = 0.175 \pm 0.007\ m_0$. Fixing this value of $m_r$, we then determine the values of $r_0$ and $\kappa$ that best model both the energies and the field-dependent shifts of the more tightly bound exciton states at lower fields. The influence of these two parameters is not completely independent—both affect the separation between the $ns$ exciton energy levels. However, while $\kappa$ affects all the exciton energies (similar to bulk semiconductors), $r_0$ influences primarily the lowest (smallest) $1s$ exciton state and therefore mostly tunes only the calculated $1s$–$2s$ energy spacing. This puts significant constraints on the values of these material parameters. For our hBN-encapsulated WS$_2$ monolayer, we find that $r_0 = 3.4 \pm 0.1$ nm and $\kappa = 4.35 \pm 0.10$ reproduces the energies and field-dependent shifts of all the experimental data very well, as shown in Fig. 1e.

As discussed in recent literature[39–41,45,46,50–54], the dielectric environment near a monolayer TMD significantly influences both the exciton binding energy and the free-particle bandgap energy. Such "Coulomb engineering" could be used to tune and laterally modulate the band structure of 2D semiconductors using (patterned) dielectrics, with hBN being an obvious choice of dielectric material. From the data and model shown in Fig. 1e, we directly determine that for hBN-encapsulated monolayer WS$_2$ the free-particle bandgap is $E_{gap} = 2.238 \pm 0.003$ eV, the binding energy of the $1s$ exciton ground state is $180 \pm 3$ meV, and its rms radius $r_{1s} = 1.8$ nm. These fundamental material parameters, which we anticipate will be of use for the future design of van der Waals heterostructures incorporating hBN and WS$_2$ monolayers, are shown in Table 1. Due to hBN encapsulation and resultant dielectric screening, the binding energy and free-particle gap are smaller than observed in non-encapsulated monolayers[31].

Figure 1e also confirms that the $1s$:$2s$:$3s$ ratios of exciton binding energies in hBN-encapsulated WS$_2$ ($1:\frac{1}{4.7}:\frac{1}{11.3}$) deviates markedly from the expectations of a purely hydrogenic $-1/r$ potential in 2D ($1:\frac{1}{9}:\frac{1}{25}$) or in 3D ($1:\frac{1}{4}:\frac{1}{9}$), an expected consequence of the non-hydrogenic electrostatic potential that exists in real 2D materials. In fact the measured binding energy ratios correspond more closely to expectations from a 3D hydrogenic potential—a consequence of the relatively large dielectric constant of the encapsulating hBN slabs used here. We note that a suitably modified hydrogenic potential can give analytic solutions and reasonable agreement with the measured exciton energies in hBN-encapsulated WSe$_2$ monolayers[55].

The inferred value of $\kappa$ is in quite reasonable agreement with the reported high-frequency dielectric constant of hBN ($\varepsilon_{hBN} \approx 4.5$)[56]; however we point out that the use of a fixed $\kappa$ is likely oversimplified, since $\varepsilon_{hBN}$ is known to vary at energies below ~100 meV due to optical phonon modes, and therefore the effective $\kappa$ may in fact vary somewhat for different $ns$ exciton states.

Finally, we note that the experimentally determined exciton mass in monolayer WS$_2$ ($m_r = 0.175\ m_0$) is slightly heavier (by ~10%) than predicted by recent density functional theory[10–12], and that the monolayer's screening length $r_0 = 3.4$ nm is about 10% smaller. A discrepancy of similar magnitude was also recently reported for the exciton mass in monolayer WSe$_2$ (ref. [27]), suggesting that further refinements of existing theories may be warranted. A possible origin of such discrepancy may be a polaron-related effect[57–60], which can give a larger correction to the exciton mass for more highly excited excitons (because their size exceeds the phonon–polaron radius and their binding energy is less than the phonon energy). If refined theories confirm such a scenario, our measurements may provide a way to estimate this polaron effect. As shown below, these differences are even more pronounced in all the molybdenum-based TMD monolayers.

**Monolayer MoS$_2$.** Following a similar measurement and analysis protocol, we next investigate the high-field magneto-absorption of MoS$_2$ monolayers encapsulated by hBN. MoS$_2$ is arguably the most well-known and archetypal member of the TMD semiconductor family, due to its abundance in the form of the natural mineral molybdenite. Consequently, MoS$_2$ was the first to be thinned to a single monolayer[3,4,61], the first to evince valley-selective optical pumping[62–65], and the first for which the electronic and exciton structure were theoretically calculated[10,11,66–72]. Despite these milestones the optical quality of monolayer MoS$_2$ was, until very recently, typically worse than its W- and Se-based counterparts, with exciton absorption and PL spectra exhibiting significant inhomogeneous broadening. Nonetheless, magneto-optical studies of the valley Zeeman effect in monolayer MoS$_2$ were first performed via reflectivity to 65 T by Stier et al.[31] and then by Mitioglu et al.[73]; however the much smaller exciton diamagnetic shifts were not measurable due to the limited optical quality of these unprotected and CVD-grown monolayers. Very recently, however, encapsulation of exfoliated MoS$_2$ monolayers by hBN was shown to significantly improve its optical quality[19], to the point where the neutral A exciton's linewidth approached the narrow homogeneously broadened limit (~2 meV) and—crucially—where excited $2s$ and $3s$ exciton Rydberg states were clearly observed in zero-field optical spectra[24]. To our knowledge, however, magneto-optical spectroscopy of the excited exciton states in monolayer MoS$_2$ has not yet been reported, and therefore an experimental determination of the exciton's mass based on diamagnetic shifts remains unavailable. An important goal of this work is to resolve this issue.

Figure 2a shows transmission spectra through an hBN-encapsulated MoS$_2$ monolayer at 0, 25, 50, and 75 T. At zero field, the spectra show the strong and narrow (width $\approx 10$ meV) absorption resonance of the ground ($1s$) state of the neutral A exciton at 1.939 eV, as well as a broader and weaker absorption ~150 meV higher in energy that is associated with ground state of the neutral B exciton, in line with previous studies and the known spin–orbit splitting of the valence bands at the $K/K'$ points of the MoS$_2$ Brillouin zone[10]. More importantly, just above the B exciton an additional narrow absorption feature appears at 2.109 eV, or ~170 meV above the A:$1s$ state. This new peak was observed recently at zero magnetic field by Robert et al.[24] in reflectivity spectra, where it was tentatively ascribed to the excited A:$2s$ state based on its energy and polarization properties. As shown immediately below, using high-magnetic field spectroscopy we confirm the identity of this $2s$ exciton, and also reveal additional highly excited $ns$ exciton Rydberg states in monolayer MoS$_2$ that can be used to determine several important material parameters, including the exciton mass $m_r$ and dielectric screening length $r_0$.

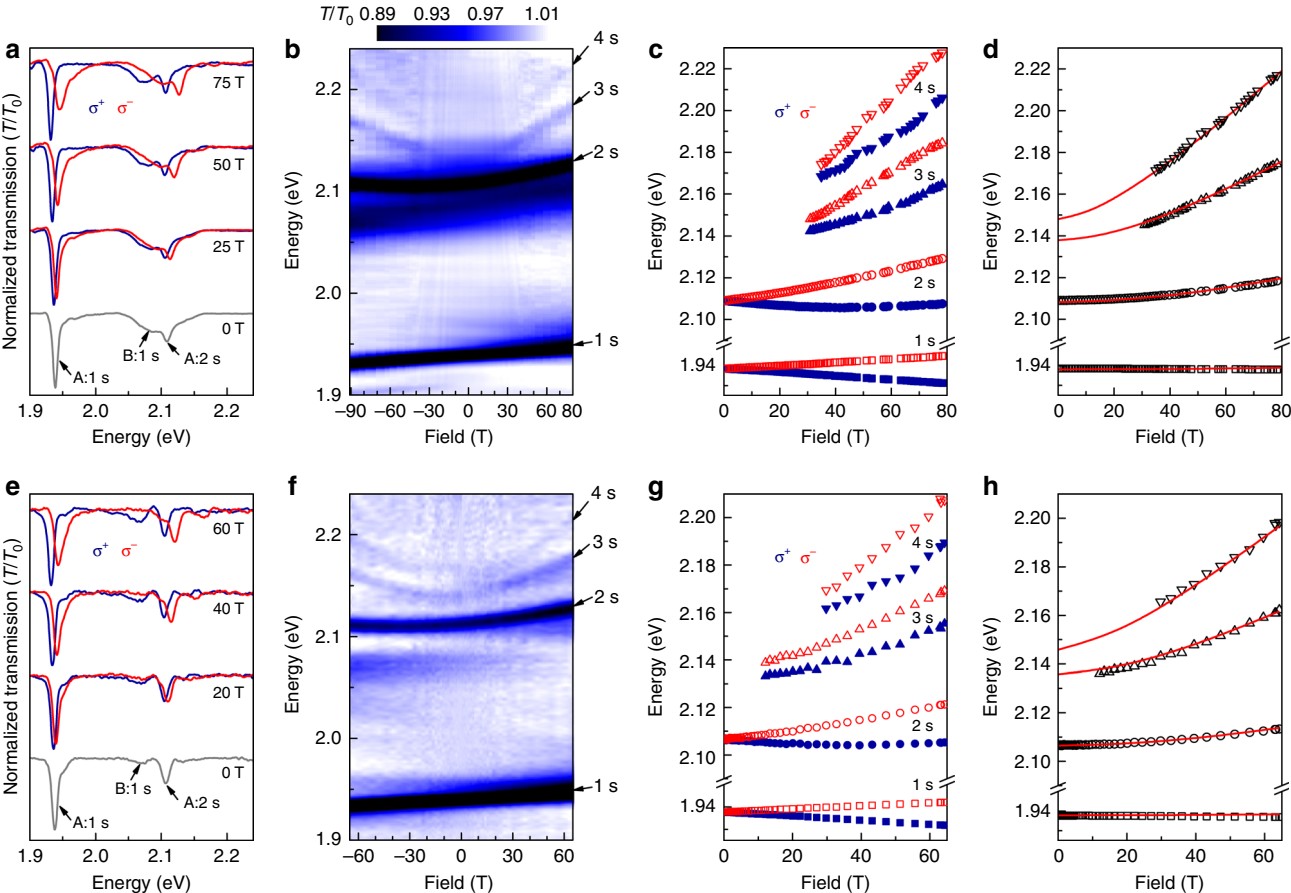

**Fig. 2** Magneto-optical spectroscopy of monolayer MoS$_2$. **a** Normalized $\sigma^{\pm}$ polarized transmission spectra through monolayer MoS$_2$ encapsulated in hBN (sample 1), at 0, 25, 50, and 75 T. **b** Intensity map showing all the transmission spectra from $-91$ T to $+80$ T. The excited 2s, 3s, and 4s Rydberg states of the neutral A exciton are visible. **c** Energies of the ns excitons for both $\sigma^{\pm}$ polarizations. **d** The average energies of the $\sigma^{\pm}$ transitions for each ns exciton state. Solid curves show model calculations ($m_r = 0.27\ m_0$, $r_0 = 3.4$ nm, $\kappa = 4.4$, and $E_{gap} = 2.161$ eV). **e–h** Data from a different hBN/MoS$_2$/hBN structure (sample 2), acquired to $\pm 65$ T. In this structure the B:1s exciton is less pronounced, and the A:4s exciton is more clear. Best fits are obtained using very similar parameter values ($m_r = 0.28\ m_0$, $r_0 = 3.4$ nm, $\kappa = 4.5$, and $E_{gap} = 2.158$ eV)

With increasing magnetic field, all the exciton resonances split and shift as shown in Fig. 2a, and additional absorption features appear at even higher energies as $|B| \rightarrow 91$ T. Again, these field-dependent trends can be identified on the intensity map shown in Fig. 2b. Besides the 1s and 2s excitons, the 3s exciton state is also clearly revealed, and a faint 4s exciton state can also be observed. The polarization-resolved $\sigma^{\pm}$ energies of these exciton states are plotted in Fig. 2c, while their polarization-averaged energies are shown in Fig. 2d. As expected, each ns state exhibits a very distinct diamagnetic shift.

As we did for the previous case of monolayer WS$_2$, we model and fit the ns exciton energies by numerically solving Schrödinger's equation using the Rytova–Keldysh potential $V_{RK}(r)$. Best results (red lines) are obtained using $m_r = 0.27 \pm 0.01\ m_0$, $r_0 = 3.4 \pm 0.1$ nm, and $\kappa = 4.4 \pm 0.1$; these values most closely reproduce the high- and low-field diamagnetic shifts of the ns exciton states, as well as their energy separations. These fits also reveal the free-particle bandgap of hBN-encapsulated MoS$_2$ monolayers ($E_{gap} = 2.161 \pm 0.003$ eV), as well as the A:1s exciton's binding energy ($E_b = 222 \pm 3$ meV).

To demonstrate the reproducibility of the samples and the reliability of our approach, Fig. 2e–h shows data from a different hBN/MoS$_2$/hBN structure that was studied to $\pm 65$ T. This sample exhibits a less pronounced B:1s exciton absorption and also a slightly stronger A:4s peak, making it easier to track the diamagnetic shifts of the ns exciton Rydberg states. Best fits to the

data yield nearly identical values for the material parameters: $m_r = 0.28 \pm 0.01\ m_0$, $r_0 = 3.4 \pm 0.1$ nm, $\kappa = 4.5 \pm 0.1$, $E_{gap} = 2.158 \pm 0.003$ eV, and $E_b = 220 \pm 3$ meV.

The experimentally determined value of the exciton's reduced mass in monolayer MoS$_2$ ($m_r/m_0 \approx 0.275$) is noticeably heavier than anticipated by leading density functional theories, where $m_r/m_0 \approx 0.24 - 0.25$ was predicted[10,11]. Although knowledge of $m_r$ does not allow to determine the electron and hole masses $m_e$ and $m_h$ separately, it is noteworthy that our result is consistent with the unexpectedly large electron mass ($m_e \approx 0.7\ m_0$) that was very recently inferred from the temperature dependence of Shubnikov-de Haas oscillations in transport studies of n-type MoS$_2$ monolayers by Pisoni et al.[28]. Given that angle-resolved photoemission spectroscopy (ARPES)[74] provides a direct measurement of the hole mass ($m_h = 0.43/0.48\ m_0$ for suspended/supported MoS$_2$), our measured value of $m_r$ implies $m_e = 0.70 \pm 0.06\ m_0$, in reasonable correspondence with the recent transport results, and well in excess (by $\sim$60%) of the electron mass calculated in recent models[10]. This suggests that a surprisingly heavy electron mass may be an intrinsic material property, and not the result of electron–electron interactions in a dense electron gas. Indeed, our magneto-transmission measurements are performed for very low exciton densities and in a regime of charge neutrality, in contrast to transport measurements where the electron density is necessarily much higher. As suggested above for the case of WS$_2$, heavier-than-expected

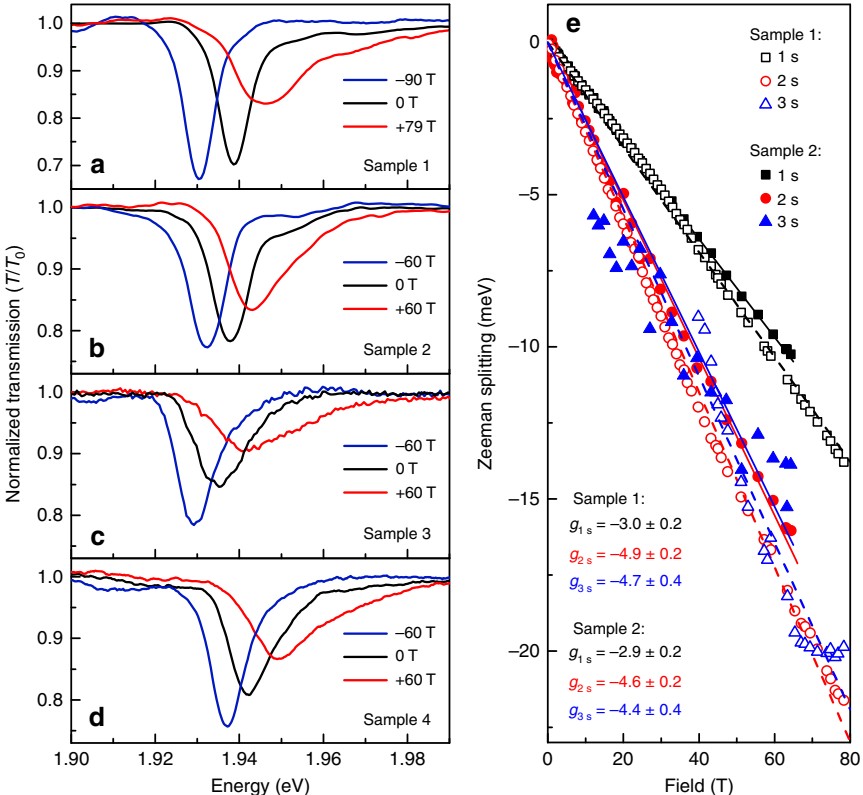

**Fig. 3** Anomalous behavior of the 1s exciton state in monolayer $MoS_2$. **a–d** Normalized transmission spectra in the vicinity of the ground state (A:1s) exciton, in four different hBN-encapsulated monolayer $MoS_2$ structures. Black curves show spectra at $B=0$ T, blue and red curves show the $\sigma^+$ and $\sigma^-$ spectra acquired at large negative and positive $B$, respectively. Note the field-induced broadening of the higher energy Zeeman state in all cases. **e** The valley Zeeman splitting of the $ns$ exciton states for samples 1 and 2; dashed lines show linear fits. The 1s state exhibits an unexpectedly small valley Zeeman splitting in comparison with the excited Rydberg states

masses may originate from polaron effects, which are predicted to give enhanced corrections in monolayer $MoS_2$ as compared to $WS_2$ (ref. [59]).

The high-field optical spectra of monolayer $MoS_2$ also reveal two unexpected, but potentially related, anomalous behaviors. The first is that the 1s exciton absorption resonance broadens significantly and asymmetrically at large $B$, but only in the higher energy $\sigma^-$ polarization. This behavior is systematic and is observed in every hBN/$MoS_2$/hBN structure that we studied. Figure 3 shows high-field spectra from four different samples. As $B$ increases and the $\sigma^-$ absorption peak undergoes a Zeeman shift to higher energies, the asymmetric broadening appears to evolve from the weak shoulder that lies ~15 meV above the 1s absorption resonance at $B=0$ (the same shoulder was also observed in the zero-field reflectivity studies of hBN/$MoS_2$/hBN by Robert et al.[24]). The nature of the state that gives rise to this shoulder is not yet understood; nevertheless large magnetic fields could force a crossing between it and the blue-shifting ($\sigma^-$) branch of the 1s bright exciton. If these states mix in large $B$, an apparent asymmetric broadening of the $\sigma^-$ polarized bright state can be expected.

Due to this asymmetric broadening and field-dependent lineshape, a very precise and reproducible fitting of the 1s exciton energy is difficult and susceptible to small but systematic deviations that depend on the exact form of the fitting function used. Consequently, we can report only that the measured diamagnetic shift of the 1s exciton is very small, of order 0.1 $\mu$eV $T^{-2}$. This is consistent with the exciton's large mass, small radius, and small diamagnetic shift that we determine by modeling the $ns$ exciton states as described above: using $V_{RK}$ and

the material parameters determined above, we calculate that $\sigma_{1s} = 0.12$ $\mu$eV $T^{-2}$, which can be in turn used to estimate the A:1s exciton's rms radius in encapsulated $MoS_2$ monolayers ($r_{1s} = 1.2$ nm).

Fortunately, however, the broadening and asymmetry of the 1s exciton does not preclude a robust determination of the much larger Zeeman splitting, which highlights the second anomalous behavior revealed by the high-field magneto-optical data: The valley Zeeman splitting of the A:1s ground state exciton in hBN-encapsulated $MoS_2$ monolayers is unexpectedly small. As shown in Fig. 3e the 1s exciton splits linearly with field, but with a small effective g-factor of only $g_{1s} \approx -3.0$. In contrast, the excited 2s, 3s, and 4s Rydberg excitons all exhibit larger g-factors from $-4.4$ to $-4.9$. Note that an anomalously small value of $g$ for only the 1s exciton state is at odds with results from monolayer $WS_2$ (Fig. 1e) and from monolayer $WSe_2$ (ref. [27]), for which all the $ns$ exciton states exhibited very similar Zeeman splittings with $g \approx -4$.

Based on our measurements of multiple hBN/$MoS_2$/hBN structures, we also note that the anomalous Zeeman splitting of the 1s exciton does appear to depend systematically on the overall optical quality of the sample, with our highest-quality structures (samples 1 and 2) showing $g_{1s} \approx -3.0$, while lesser-quality structures that exhibit broader absorption lines and fewer $ns$ states show $g_{1s} = -3.6$ and $-3.8$ (samples 3 and 4, respectively). These trends, though not yet understood, are nonetheless very consistent with previously reported studies: Lower quality unprotected $MoS_2$ monolayers with broad 1s absorption lines exhibited $g_{1s} \sim -4$ (refs. [31,73]), while extremely high-quality hBN-encapsulated $MoS_2$ monolayers with very narrow exciton linewidths[19] exhibited a very small $g_{1s} = -1.7$. We tentatively

suggest that the small valley splitting of the A:1s exciton in hBN/MoS$_2$/hBN may also result from the close proximity to and interaction with the nearby absorption feature that lies 15 meV above the exciton (e.g., an avoided crossing would suppress the Zeeman shift of the $\sigma^-$ branch of the bright exciton). Although the origin of this higher-lying state is not yet known, one possibility is that it may be related to nominally spin- and valley-forbidden "dark" or "gray" exciton states[75–78]. Whereas in monolayer WSe$_2$ these dark states are known to lie far (~40 meV) below the optically allowed exciton states due mainly to the large spin–orbit splitting of the conduction bands, in monolayer MoS$_2$ the conduction band spin–orbit splitting is thought to be much smaller and of opposite sign, such that dark/gray states may actually lie slightly above the bright states[75].

**Monolayer MoSe$_2$.** Next, we perform high-field magneto-absorption of hBN-encapsulated MoSe$_2$ monolayers. Together with WSe$_2$, MoSe$_2$ was the first monolayer TMD semiconductor to be measured in a magnetic field where, in modest fields (<10 T), the Zeeman splitting of the A:1s exciton ground state was observed by polarized PL[79–81]. Similar valley Zeeman splittings were subsequently reported to 65 T on CVD-grown MoSe$_2$ monolayers[73]. However, in all these magneto-optical studies the much smaller exciton diamagnetic shift was not detected, likely due to the limited optical quality of the samples and the (theoretically predicted) heavier exciton mass[10,11]. In fact, only very recently was the optical quality of monolayer MoSe$_2$ improved—again by hBN encapsulation—to the point where excited A:2s excitons were observed at zero field, for example by Han et al.[22] and by Horng et al.[23]. However, high-field spectroscopy of excited ns excitons has not been reported to date, and consequently an experimental determination of the exciton mass in monolayer MoSe$_2$ is still lacking, along with other fundamental parameters.

Figure 4a shows normalized magneto-transmission spectra from a MoSe$_2$ monolayer encapsulated by hBN, at 0, 20, 40, and 60 T. The zero-field spectrum shows a very sharp (width ≈6 meV) A:1s exciton ground state at 1.643 eV. A broader absorption located ~200 meV higher in energy corresponds to the B:1s exciton, in line with past work[22] and the expected spin–orbit splitting of the valence bands at the $K/K'$ points of the Brillouin zone[10]. A weak absorption feature located at 1.811 eV, just below the B:1s exciton, corresponds closely to a similar weak feature that was reported recently by Han et al.[22], which was tentatively ascribed to the excited A:2s exciton based on reflectivity and upconversion measurements. This means that even higher A:3s and A:4s Rydberg states in monolayer MoSe$_2$ directly overlap with the broad absorption of the B:1s exciton, making their presence difficult to detect, at least at zero applied field.

With increasing field to 65 T, the A:1s exciton shows a very obvious Zeeman splitting (Fig. 4a, b). The weaker state tentatively ascribed to the A:2s exciton at ~1.811 eV is difficult to follow as it shifts and merges with the B:1s exciton. Interestingly, however, at very large fields above 40 T the spectra show two new resonances emerging rapidly from the high-energy side of the broad B:1s absorption. These features can also be readily identified in the detailed fits shown in Fig. 4c. As before, these field-dependent trends can be seen in the intensity plot of Fig. 4b. The energies of all the observed absorption features are plotted in Fig. 4d, and the polarization-averaged energy of each of these states are plotted in Fig. 4e.

Fitting these data to the numerically computed ns exciton energies (again using the Rytova–Keldysh potential for a 2D material) strongly suggests that the two highest-energy absorption features are the A:3s and A:4s excited Rydberg excitons. Using a large reduced mass of $m_r = 0.350 \pm 0.015\ m_0$ and material parameters $r_0 = 3.9 \pm 0.1$ nm and $\kappa = 4.4 \pm 0.1$, we find that

the calculated energies and field-dependent shifts of the $1s - 4s$ exciton states (red lines) accurately match the measured energy separations and field-dependent diamagnetic shifts of the experimental data. In particular the very small quadratic diamagnetic shift of the A:1s exciton ($\sigma_{1s} = 0.07\ \mu$eV T$^{-2}$) is captured very well, from which we also determine its small rms radius, $r_{1s} = 1.1$ nm. In addition, the model also directly reveals the binding energy of the A:1s exciton ($231 \pm 3$ meV), and the free-particle bandgap in hBN-encapsulated MoSe$_2$ monolayers ($E_{gap} = 1.874 \pm 0.003$ eV).

Similar to the case of monolayer MoS$_2$, the exciton mass that we measure in monolayer MoSe$_2$ ($m_r = 0.35\ m_0$) is substantially larger than anticipated by current density functional theories, which predict $m_r \approx 0.27\ m_0$[10,11]. Again we note that the large measured value of $m_r$ is in fact qualitatively consistent with the unexpectedly large electron mass, $m_e \approx 0.8\ m_0$, that was recently suggested by transport studies of $n$-type MoSe$_2$ monolayers by Larentis et al.[29] Taking $m_h = 0.6 \pm 0.1\ m_0$ from ARPES measurements[82–84], our value of $m_r$ also yields a surprisingly heavy $m_e \approx 0.84\ m_0$ in monolayer MoSe$_2$, which is about 70% more than recent DFT calculations[10]. Similar to the case of monolayer MoS$_2$, this suggests that the unexpectedly large $m_e$ may result not from interactions with other electrons, but may be an intrinsic material property of MoSe$_2$ monolayers.

The spectra also reveal approximately similar Zeeman splittings for all the measurable exciton states in monolayer MoSe$_2$ (Fig. 4f). For the 1s ground state, we find $g_{1s} = -4.3$, consistent with previous reports[73,79–81], while for the 3s and 4s excited states, we find somewhat larger values of about $-4.9$. Finally, in these MoSe$_2$ monolayers the excited B:2s exciton can also be observed; its Zeeman splitting and diamagnetic shift are discussed in the Supplementary Information.

**Monolayer MoTe$_2$.** Finally, we measure hBN-encapsulated MoTe$_2$ monolayers. Of all the monolayer TMD semiconductors, the exciton properties of MoTe$_2$ remain among the least well explored. The Zeeman splitting of the A:1s ground state was first measured to 30 T by Arora et al.[85], and the A:2s excited state was later identified by Han et al.[22] via reflection and PL upconversion spectroscopy. To date, however, no diamagnetic shifts have been experimentally measured and therefore the exciton mass and other fundamental parameters have not yet been determined.

Figure 5a shows an intensity plot of the magneto-transmission through a MoTe$_2$ monolayer encapsulated by hBN. At B=0, the spectrum shows a sharp (width ≈6 meV) A:1s exciton ground state at 1.175 eV and a weaker A:2s absorption feature at 1.299 eV, in close agreement with prior zero-field studies[22].

In applied field, both exciton features exhibit a similar Zeeman splitting (Fig. 5b, d) and clear quadratic blueshifts (Fig. 5c). The measured diamagnetic shifts are: $\sigma_{1s} = 0.10\ \mu$eV T$^{-2}$ and $\sigma_{2s} = 1.4\ \mu$eV T$^{-2}$. Unfortunately (and in contrast to the other TMD monolayers), higher-lying Rydberg excitons are not observed even in large field, which prevents an especially precise fitting to the calculated ns exciton energies. However, using the A:1s and A:2s diamagnetic shifts and their energy separation, and also assuming that $\kappa = 4.4$ (the average value for all other hBN-encapsulated monolayers), we are nonetheless able to estimate the remaining parameters of the model. Using $m_r = 0.36 \pm 0.04\ m_0$ and $r_0 = 6.4 \pm 0.3$ nm, the calculated energies and shifts accurately match the data. From the model we also infer the small A:1s radius ($r_{1s} = 1.3$ nm) and binding energy ($177 \pm 3$ meV), as well as the free-particle bandgap in hBN-encapsulated monolayer MoTe$_2$ ($E_{gap} = 1.352 \pm 0.003$ eV). The inferred mass is once again significantly heavier (by 25%) than

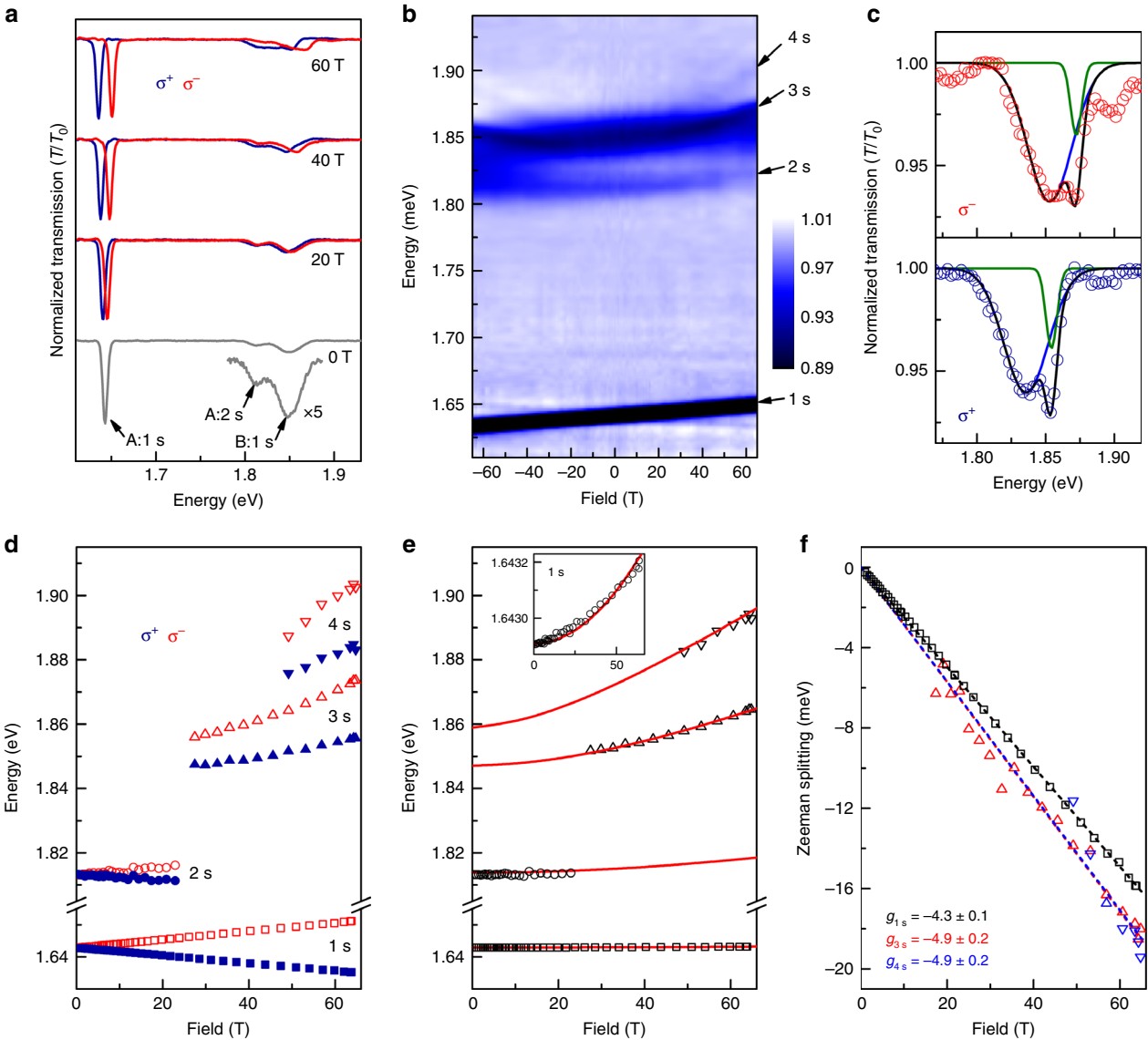

**Fig. 4** Magneto-optical spectroscopy of monolayer MoSe₂. **a** Normalized transmission spectra through an hBN-encapsulated MoSe₂ monolayer at selected magnetic fields, for both $\sigma^{\pm}$ polarizations. The inset shows a 5× magnified spectrum of the higher energy features at 0 T. **b** Intensity map showing all the spectra from −65 to +65 T. Excited $ns$ Rydberg states of the neutral A exciton overlap with and emerge from the broader absorption at 1.85 eV that is due to the B exciton ground state. **c** Examples of two-Gaussian fits to the broad feature near 1.85 eV in $\sigma^{-}$ (upper panel) and $\sigma^{+}$ (lower panel) polarization at 65 T. In each panel, the two overlapping absorption peaks correspond to the B:1$s$ exciton (wide blue curve) and the A:3$s$ exciton (narrow green curve). The A:4$s$ state is also visible at ~1.9 eV. **d** Measured exciton energies for both $\sigma^{\pm}$ polarizations. **e** The average energies of the $\sigma^{\pm}$ transitions. Red lines show the modeled exciton energies (see text). Parameters: $m_r = 0.35\ m_0$, $r_0 = 3.9$ nm, $\kappa = 4.4$, and $E_{gap} = 1.874$ eV. The states that emerge at high magnetic fields from the B:1$s$ exciton absorption correspond very well to the A:3$s$ and A:4$s$ Rydberg states. Inset: Expanded plot of the 1$s$ exciton energy, showing its very small quadratic diamagnetic shift. **f** The Zeeman splitting of the A:1$s$, A:3$s$, and A:4$s$ exciton states; dashed lines show linear fits

theoretical expectations[10]. It is also the heaviest of all the Mo-based TMD monolayers, which confirms the predicted general trend of increasing exciton mass with increasing chalcogen atomic mass.

## Discussion

Magneto-absorption spectroscopy of high-quality hBN-encapsulated MoS₂, MoSe₂, MoTe₂, and WS₂ monolayers allows to resolve and follow the field-dependent diamagnetic shifts and splittings of not only the 1$s$ (ground state) excitons, but also the excited $ns$ Rydberg excitons. This permits a detailed analysis and fitting of the data to experimentally determine a number of essential material parameters for monolayer TMD

semiconductors including the exciton's reduced mass $m_r$, the binding energies and rms radii of the various $ns$ exciton states, the free-particle bandgap $E_{gap}$, and the monolayer's dielectric screening length $r_0$. These fundamental parameters are listed in Table 1. These results experimentally confirm long-standing expectations of heavier exciton masses and larger dielectric screening lengths as the chalcogen atomic mass increases (from S to Se to Te), and also as the metal atom becomes lighter (from W to Mo). It is anticipated that these experimentally determined material parameters will prove useful for the rational design and engineering of future optoelectronic van der Waals hetero-structures that incorporate TMD monolayers and hBN.

It is noteworthy that the measured reduced masses $m_r$ are consistently and substantially larger than anticipated by modern

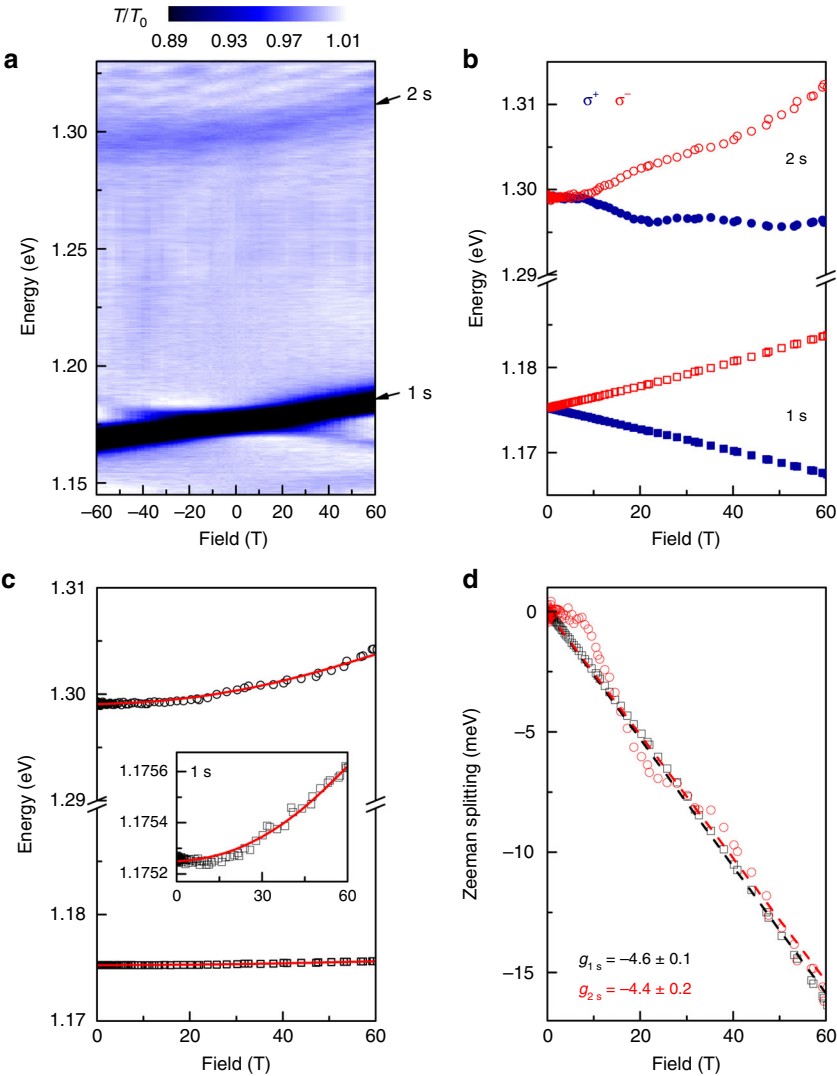

**Fig. 5** Magneto-optical spectroscopy of monolayer $MoTe_2$. **a** Normalized intensity map of the field-dependent transmission spectra through monolayer $MoTe_2$, showing the A:1s ground state exciton and the A:2s excited state. Due to a small (<5 %) polarization leakage, a weak trace of the oppositely polarized 1s state is also visible. **b** Energies of the excitons in $\sigma^{\pm}$ polarization. **c** Averaged exciton energies showing diamagnetic shifts. Red lines show calculated energies (see text). Parameters: $m_r = 0.36\ m_0$, $r_0 = 6.4$ nm, $\kappa = 4.4$, and $E_{gap} = 1.352$ eV. Inset: Expanded plot of the 1s exciton energy, showing its very small quadratic diamagnetic shift. **d** The valley Zeeman splitting of the 1s and 2s excitons; dashed lines show linear fits

density functional theories, especially across the entire family of Mo-based TMD monolayers. In addition, together with available ARPES data on hole masses in $MoS_2$ and $MoSe_2$ monolayers, our measurements point to surprisingly large electron masses in these materials (about 60–70% larger than predicted). In this context it is also noteworthy that two very recent transport studies[28,29] have also revealed unexpectedly heavy electron masses in $n$-type $MoS_2$ and $MoSe_2$ monolayers. This suggests that a key ingredient is missing in current DFT calculations. The mass increase revealed by our measurements could for instance be explained by an electron–phonon polaron coupling[57–59]. The possible role of interface polarons in TMD/hBN structures should also be investigated[86].

## Methods

**Experimental setup**. The fiber/sample assembly is mounted in 4 K exchange gas in the tail of a liquid helium cryostat located in the bore of a pulsed magnet. We use both capacitor-driven 65 T pulsed magnets, as well as a unique generator/capacitor-driven magnet capable of ultrahigh fields up to 100 T[87]. Broadband transmission spectroscopy is performed by coupling unpolarized white light from a Xe lamp into the single-mode fiber. After passing through the sample and a thin-film circular polarizer, the transmitted light is retro-reflected back

through a separate multimode collection fiber, and is detected using a spectrometer and high-speed CCD camera. Spectra are continuously acquired every 1 ms throughout the magnet pulse. Access to both $\sigma^+$ and $\sigma^-$ circularly polarized transitions (corresponding to interband optical transitions in the $K$ and $K'$ valley of the TMD monolayer, respectively) is achieved by reversing the direction of the magnetic field.

## Data availability

All data generated and analyzed during this study are included in this published article (and its supplementary information files).

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

## Acknowledgements

We gratefully acknowledge helpful discussions with H. Dery, K. Velizhanin, M. Potemski, and M. M. Glazov, and technical support with the 100 T magnet from H. Teshima, D. Roybal, M. Pacheco, D. Nguyen, R. McDonald, J. Martin, M. Hinrichs, F. Balakirev, and J. Betts. Work at the NHMFL was supported by the Los Alamos LDRD program and the DOE BES 'Science of 100 T' program. The NHMFL is supported by the National Science Foundation DMR-1644779, the State of Florida, and the U.S. Department of Energy. We acknowledge funding from ANR 2D-vdW-Spin, ANR VallEx, Labex NEXT projects VWspin and MILO, ITN Spin-NANO Marie Sklodowska-Curie grant agreement No 676108 and ITN 4PHOTON No 721394. S.A.C. acknowledges an invited researcher grant from Toulouse Labex NEXT. X.M. acknowledges the Institut Universitaire de France. Growth of hBN crystals was supported by the Elemental Strategy Initiative conducted by the MEXT, Japan and the CREST (JPMJCR15F3), JST.

## Author contributions

S.A.C., B.U. and X.M. conceived the idea and supervised the research. C.R., S.S, E.C and S.A.C. fabricated the samples. T.T. and K.W. synthesized the hBN. M.G., J.L., A.V.S and C.R. performed the magneto-optical measurements. M.G. and J.L. performed data analysis. All authors discussed the results. M.G., J.L. and S.A.C. wrote the manuscript in consultation with all authors.

## Additional information

**Competing interests:** The authors declare no competing interests.

