## [Peer Review File · Nature Communications]

Reviewers' comments:

Reviewer #1 (Remarks to the Author):

I had an opportunity to carefully read the manuscript by Goryca et al. about magneto-optical spectroscopy of WS₂ and MoSe₂ up to 65 T, and MoS₂ up to 90T magnetic fields. The main result of the paper is to find the effective masses of the excitons by performing magneto-optics of Rydberg states. The authors compare the experimentally derived effective masses with those calculated using DFT in literature and find discrepancies up to 20%. The paper has new results, e.g. the effective exciton masses in the monolayers of these materials have been derived for the first time experimentally. The physics of the paper has already been well established in multiple works e.g. (Ref. 27, and references therein, <https://arxiv.org/abs/1902.03962v1> and 10.1103/PhysRevB.99.205420, Ref. 31, Ref. 67). I feel that the present work lacks sufficient novelty for Nature Communications, as well as lacks in covering a general readership. The work should be submitted to a more specialized journal.

I have a few comments on the paper:

1) The main result of the paper is that the experimentally derived effective masses differ 'significantly' (up to 20%) than the calculations in literature cited. However, a quick look at table V of PHYSICAL REVIEW B96, 075431 (2017) suggests that the DFT theories provide a large spread of electron and hole masses (e.g. a factor of 2 in many cases in this table). Therefore, I am not convinced if the conclusions derived by the authors are solid.

2) The authors cite multiple papers where the electronic and hole masses are routinely determined from transport measurements (e.g. Refs. 28, 29 and PRL 116, 086601 (2016) for hole mass in WSe₂ etc.). Furthermore, using magneto-optics under high magnetic fields, similar results are available in Ref. 27. I find that the present work does not contribute much to the new physics.

3) The physics of encapsulation by hBN and the excited exciton states, where binding energies are derived has already been discussed in details up to 30 T e.g. <https://arxiv.org/abs/1902.03962v1> and 10.1103/PhysRevB.99.205420

Other minor comments: A) Is the formula for diamagnetic shift applicable to 2D materials? or is it a 3D formula?

B) How is the rms radius defined for the exciton? Does it take into account the our of plane confinement of the exciton?

Reviewer #2 (Remarks to the Author):

In this paper the authors report an optical spectroscopy study, in extremely high magnetic fields, of excitons in MoS₂, MoSe₂, and WS₂ monolayers. They follow the diamagnetic shift of the series of exciton Rydberg states to extract accurately and in a quite consistent manner the effective masses and binding energies of excitons (and some other material parameters). The authors have performed an excellent experimental analysis, with a massive amount of careful measurements. The data analysis is sound. It relies on the well understood behavior of excitons in a magnetic field and on the broadly adopted model of the confined potential (Rytova-Keldysh model) to model the spectrum of excitonic excited states.

The key results are summarized in Table 1, where one could find the most fundamental parameters of excitons in four archetypical direct-gap monolayer semiconductors - MoS₂, MoSe₂, WS₂, and WSe₂ (for this material, the authors cite the results from their previously reported study, Ref. [27]). These materials are the foundational blocks for the rapidly expanding field of TMDC-based heterostructures. Thus, the reported results are very timely and will certainly be of interest for the broad community of researchers working on TMDCs. However, I have several questions and remarks, which I ask the authors to address before I recommend the results to be published in Nature Communications:

(1) This study reports the band gap for WS₂ (MoS₂) to be about 2.238 eV (2.160 eV) with the accuracy of +/- 3 meV. In 2016, the authors published in Nature Communications a similar study of WS₂ and MoS₂ monolayers on Si/SiO₂. For WS₂, they reported 410 meV binding energy and ~ 2.044 eV exciton energy. This would give the bandgap at about 2.45 eV, i.e. 200 meV higher compared to the value reported here. Similar discrepancy can be found for MoS₂. Of course, the TMDC monolayers on hBN and on SiO₂ are exposed to different dielectric environment, so the binding energy could be different. However, this should not affect the value of the single-particle band gap. Since the authors employed similar experimental methods and analyzed the data within the same Rytova-Keldysh model, I would expect to see in this manuscript a discussion comparing these two studies.

(2) On pages 7 (last paragraph) and 8, the manuscript shows and discusses two unexpected properties of the 1s exciton in monolayer MoS₂ - asymmetric broadening at high field and the small valley Zeeman g-factor. The authors suggested that the high-energy shoulder lying at ~15 meV above the main peak could be the same feature observed in zero-field spectra in Ref. 24 and its origins could be related to dark/grey excitons. I don't think the data presented in this manuscript or in Ref. 24 allow to draw such conclusions. Dark excitons should have the g-factor two times or more larger than the bright one. If the dark exciton is at 15 meV above the bright exciton at zero field, then it should shift away rapidly and won't be at the shoulder position observed in this work.

(3) As for the surprisingly small g-factor of the 1s state in monolayer MoS₂, the authors "tentatively suggest" that it may arise from interaction with dark excitons. Even considered as a possible mechanism, it requires some justification. Could authors discuss what/how the interaction with dark excitons would result in changing and specifically, in lowering the g-factor, and why this eventual interaction is absent or not effective in the lower quality samples.

(4) The color map in Fig. 4 seems to show that the 3s line is stronger than the 2s one (on the +65T side, in particular). This is totally unexpected if this is the case. Could the authors check the intensities of 2s and 3s lines at +65T and, of course, if the unusual ratio of 2s/3s intensities is confirmed, discuss its possible origins.

(5) A very important starting point of this study is the claim that the measurements were performed on charge neutral samples. This very well could be the case. However, a more correct statement would be that the samples were not intentionally doped. What about the unintentional doping? As far I know, it is always necessary to work with gated samples and apply the gate voltage to bring the sample to the charge neutrality point. While the high-field pulsed experiments are hardly to be performed on gated samples because of the extreme nature of the pulsed field environment, it is important to estimate the carrier density in the studied (or similar) samples. Moreover, this analysis is necessary to support the discussion about the origins of "heavier-than-expected" excitons in Mo-based monolayers, which may be an intrinsic property of their bandstructure (as proposed here) or resulted from e-e interactions at (somewhat) higher carrier density. I am wondering if the authors performed or could perform PL characterization. The presence (or absence) of charged excitons may indicate that the samples are doped (or intrinsic).

(6) Overall, I found that the manuscript is long and overloaded with somewhat technical and/or redundant information. I think moving such details to Suppl.Materials, focusing more on the physics and shortening the main text could make the manuscript better and help it meet with the high standards of Nature journals. For example, it might be enough to say in the introduction that higher order exciton states have not been previously observed in specific materials and not to repeat it in the following sessions. It may not be necessary to show the results obtained on multiple samples in Figs.2 and 3, etc.. Also, the manuscript sounds to me employing repeatedly unnecessarily subjective and dramatic vocabulary (crucially, importantly, interestingly, ...).

(7) The reference list seems to be detailed and up to date. Still, the authors may consider to add in the appropriate context the following references: Chen et al, Nano Lett. 19, 42464 (2019) - Rydberg excitons and Zeeman splitting in WSe₂; Koperski et al, 2D Mater. 6 015001 (2019).

Reviewer #3 (Remarks to the Author):

The manuscript by M. Goryca et al. provides to the community a set of material parameters such as exciton reduced mass, exciton bohr radius, g-factors for three different two-dimensional semiconductors, namely WS₂, MoS₂ and MoSe₂. These informations are extracted from delicate low temperature, polarization resolved, absorption experiments in high magnetic fields (up to 91T) provided by a pulsed resistive solenoid.

These results are obtained on high quality 2D semiconducting materials encapsulated in hexagonal boron nitride to strongly enhance the optical quality and reach the homogenous limit resulting in thin linewidths of the excitonic absorptions.

The authors use a technique that they (part of the authors) have developed and used to study monolayers of WSe₂ (reported in Ref 27) : they build the van der Waals hetrostructure directly on the surface of the optical fiber and collect the transmitted light.

In this manuscript, the authors successfully extend this technique to the other members of the 2D semiconductor family. The main results appear to be: i) the experimental determination of reduced mass for these three compounds, and the discrepancy between the measured reduced excitonic masses and the ones predicted by DFT, for Mo based compounds. The introduction and the body of the manuscript include references to a large and relevant part of the litterature. The data are of high quality, are well presented and commented.

The case of MoS₂ is of particular interest and this manuscript adds some new data concerning this topic but doesn't provide any explanation (anomalously small g-factor for the 1s state, heavier masses than expected). Even if the B exciton spectrum overlaps with A exciton excited states due to the smaller valence band splitting (also in MoSe₂), exciton excited states can be observed, allowing for the analysis of the excited state sequence through the RK potential. No special originality here compared to the study reported on WSe₂ by different groups, including some of the authors.

The sample used are encapsulated in hBN which gives in PL emission linewidth in the 1-2 meV (FWHM). Here, the authors perform absorption experiments on similar structure and show absorptions for the A:1s closer to 10 meV in FWHM. Could the authors comment on this point ? In general, could the author indicate the FWHM that they measure in absorption for A:1s, A:2s etc ... (this is complicated to do on the figures provided by the authors because of the scale) As I understand, the technique to produce the sampels doesn't allow to move the sample on the fiber after transfer which makes it difficult to select the best, "bubble free", locations.

The authors also describe the field evolution of the B:1s and 2s state in MoSe₂, observable in their high quality sample. They indicate a diamagnetic shift of $1.3 \mu\text{eV}/\text{T}^2$, significantly larger than for the A:1s. There is a problem here, probably a typo, because the text refers to the B:2s, Fig 5b shows the evolution of a peak around 2.017 eV but it is labeled B:1s.

Let's assume that this is a typo and that Fig 5b refers to B:2s, then what is the diamagnetic shift of B:1s ? how does it compare to the one of A:1s ?

For me, this is a very nice experimental work bringing to the community a set of parameters describing the electronic and optical properties of the most studied 2D semiconducting materials, but appears largely as a follow up of Ref 27, extended to other materials. I cannot find a clear and original message delivered by this paper and, to my opinion, it should be published in a more specialized journal.

Note to all reviewers:

Many thanks for the detailed reports and feedback on our manuscript. Point-by-point responses are written below, with the reviewers' original remarks in *blue italics*. We regret our delay in returning a revised version of the manuscript, but it was necessitated by the very best of reasons: In the last month we were able to obtain new and quite exciting pulsed-magnetic-field results from the 'fifth' monolayer TMD material – MoTe₂. Together with the existing results on MoS₂ and MoSe₂, this completes the entire family of Mo-based monolayer semiconductors. Crucially, general trends across this entire family are now experimentally confirmed and established (eg, increasing mass and dielectric screening length with heavier chalcogen atom, and the persistence of anomalously heavy masses for *all* Mo-based TMDs). We have incorporated these new results into the manuscript, because we firmly believe that it is better to have one complete publication that can serve as a definitive reference, rather than many lesser reports that break up the story into many pieces.

Reply to Reviewer #1:

I had an opportunity to carefully read the manuscript by Goryca et al. about magneto-optical spectroscopy of WS₂ and MoSe₂ up to 65 T, and MoS₂ up to 90T magnetic fields. The main result of the paper is to find the effective masses of the excitons by performing magneto-optics of Rydberg states. The authors compare the experimentally derived effective masses with those calculated using DFT in literature and find discrepancies up to 20%. The paper has new results, e.g. the effective exciton masses in the monolayers of these materials have been derived for the first time experimentally. The physics of the paper has already been well established in multiple works e.g. (Ref. 27, and references therein, <https://arxiv.org/abs/1902.03962v1> and 10.1103/PhysRevB.99.205420, Ref. 31, Ref. 67). I feel that the present work lacks sufficient novelty for Nature Communications, as well as lacks in covering a general readership. The work should be submitted to a more specialized journal.

Reply: The Reviewer has indeed identified the main goal and primary discovery of this work, which is to provide the *first comprehensive and quantitative picture* of the fundamental exciton and dielectric properties of the most widely-studied Mo- and W-based monolayer semiconductors – *essential optoelectronic material properties that have never been experimentally measured until now*. Given the huge and worldwide interest in these monolayer semiconductors amongst physicists, chemists, and materials scientists -- not to mention tremendous parallel interests in assembling van der Waals heterostructures of these monolayers – we strongly believe that our first determination of the fundamental mass and dielectric parameters will indeed be very well and very widely received by an extremely broad community of scientists. *These parameters are required inputs for any rational design of optoelectronic devices incorporating monolayer TMDs*. Thus we maintain our belief that Nature Communications represents the ideal venue for these new results. Moreover the additional new findings of unexpectedly heavy electron masses in Mo-based monolayers (even at charge neutrality, unlike transport studies), and the anomalously small g-factor in MoS₂, should certainly capture the attention of the many theoreticians working on density-functional theory (DFT) of these materials, and motivate improved models.

Naturally, the underlying physics of excitons in magnetic fields in general has been known from many decades of semiconductor physics; in our view this is not a valid reason to claim a lack of sufficient novelty. By analogy, the physics of spin precession/dephasing in magnetic fields has also been known from the early days of NMR, yet NMR continues to be a primary discovery tool, especially for new materials and particularly in ever-higher magnetic fields which provide increased resolution.

Finally, we note that of the papers mentioned above, PRB **99** 205420 (WSe₂, 31T) doesn't determine a mass but rather assumes values from DFT, arXiv1902.03962 (WSe₂, 14T) also doesn't determine masses, Ref. 31 (0T) assumes a mass from DFT, and Ref. 67 is a theory paper. It is not clear why the Reviewer mentions these. [Reference numbers refer to original manuscript]

I have a few comments on the paper:

1) The main result of the paper is that the experimentally derived effective masses differ 'significantly' (up to 20%) than the calculations in literature cited. However, a quick look at table V of PHYSICAL REVIEW B96, 075431 (2017) suggests that the DFT theories provide a large spread of electron and hole masses (e.g. a factor of 2 in many cases in this table). Therefore, I am not convinced if the conclusions derived by the authors are solid.

Reply: The main result of the manuscript, as stated above and as captured in Table I, is the first measurement of both exciton and dielectric parameters across the entire family of monolayer TMD semiconductors. A consequence of these measurements is to then note that the measured masses significantly exceed values determined by *modern state-of-the-art* DFT calculations (Ref. 10 by Kormanyos in particular), especially for Mo-based monolayers, in apparent agreement with very recent and exciting transport studies reporting anomalously heavy electron masses. We emphasize “state-of-the-art” because many early DFT theories of these monolayers (circa 2012) -- which are sometimes included in various tables including Table V in PRB **96** 075431 – used rather simple approaches that are known to be inaccurate. For example, results from the non-self-consistent G_0W_0 approach [PRB **86** 115409 (2012)] are included in this Table V, but even those authors note that this approach has inherent levels of approximation (and significantly overestimates the mass) as compared to more refined methods. We believe it is not especially relevant or useful to compare our results with outdated theories. The revised manuscript further emphasizes that we compare our data to current state-of-the-art models, ref. 10 in particular. And as stated in the manuscript, a 20-30% increase in reduced mass corresponds to *much larger* in electron mass, which is a significant deviation from existing theory.

2) The authors cite multiple papers where the electronic and hole masses are routinely determined from transport measurements (e.g. Refs. 28, 29 and PRL 116, 086601 (2016) for hole mass in WSe₂ etc.). Furthermore, using magneto-optics under high magnetic fields, similar results are available in Ref. 27. I find that the present work does not contribute much to the new physics.

Reply: As we tried to explain in the introduction to the manuscript, transport measurements by definition measure properties in a relatively high density electron (or hole) gas. Two of the most interesting recent transport studies of TMD materials have reported surprisingly and unexpectedly heavy electron masses in monolayer MoS₂ and MoSe₂ [Refs 28, 29]. Because they are transport measurements, however, an open question is whether the heavy masses arise from electron-electron interactions in the high-density gas, or are instead an unanticipated but intrinsic property of Mo-based monolayers. Fortunately, magneto-optics provides an alternative and quite complementary approach by allowing to measure these properties in the limit of zero doping. This provides a means to directly answer this important outstanding question. We strongly feel this represents an obvious and significant contribution to new physics in these 2D materials.

Further, we point out that Ref. 27 concerns only WSe₂, whereas the current manuscript measures the (very different) fundamental material parameters across the *entire family* of the most well-studied TMD monolayers including MoSe₂, WS₂, and the archetypal MoS₂ (and now also MoTe₂) – none of which had been measured previously. In addition this work provides both the masses *as well as* the equally-important dielectric screening properties of these monolayers (which was not measured in [27]). These are clearly not merely ‘similar results’, but rather provide a comprehensive and quantitative picture of the important optoelectronic parameters of all these 2D semiconductors, which will serve as essential inputs for the design

of real van der Waals devices in the future to the very large community of researchers in the field. [Reference numbers refer to original manuscript]

3) *The physics of encapsulation by hBN and the excited exciton states, where binding energies are derived has already been discussed in details up to 30 T e.g. <https://arxiv.org/abs/1902.03962v1> and [10.1103/PhysRevB.99.205420](https://arxiv.org/abs/10.1103/PhysRevB.99.205420).*

Reply: Yes, hBN encapsulation is a standard technique, now used by many groups including ours, by which high-optical quality monolayer samples are achieved. And further, exciton binding energies can be (and have been) inferred from magneto-optics -- as we demonstrated in 2018 for WSe2 and as other groups have subsequently also shown. *We reiterate, however, that the central focus and main discovery of our manuscript is the measurement of the most fundamental material parameters – mass and dielectric screening – across the entire family of monolayer TMD semiconductors* (the data do also happen to provide very accurate binding energies, but this is not the main message). Additional key results are the unexpectedly large electron masses in Mo-based TMDs, and also the anomalous g-factors and broadening in MoS2 that likely arise from interactions with the nearby absorption resonance.

Other minor comments: A) Is the formula for diamagnetic shift applicable to 2D materials? or is it a 3D formula?

Reply: Yes, expression for diamagnetic shift in the weak-field limit, $\Delta E_{dia} = (e^2 B^2 / 8m) \langle \psi | r^2 | \psi \rangle$, is valid for 2D materials (see, e.g., Refs [13, 14] in original manuscript). It depends only on the mass and root-mean-square radius of the exciton, which is defined explicitly below and also in the manuscript. The diamagnetic shift depends on the dimensionality of the material only insofar as dimensionality affects the exciton wavefunction ψ .

B) How is the rms radius defined for the exciton? Does it take into account the our of plane confinement of the exciton?

Reply: As described in the section “Excitons in weak- and strong-field limits” on page 4 of the main text, the rms radius is defined as $r_{ns} = \sqrt{\langle r_p^2 \rangle} = \sqrt{8m_r \sigma_{ns}} / e$, where σ_{ns} is the diamagnetic shift and r_p is a radial coordinate perpendicular to the applied field (i.e., in the 2D plane of the sample), and $\langle r_p^2 \rangle = \langle \psi_{ns}(r) | r_p^2 | \psi_{ns}(r) \rangle$ is the expectation value computed over the envelope wavefunction of the ns exciton. r_{ns} represents the characteristic in-plane electron-hole separation. The diamagnetic shift σ_{ns} is experimentally measured, and for comparison with calculations $\psi_{ns}(r)$ is modeled as being two-dimensional (effectively pancake-shaped) – a good assumption given the out-of-plane confinement of the exciton to the 2D plane of the monolayer semiconductor.

Reply to Reviewer #2:

In this paper the authors report an optical spectroscopy study, in extremely high magnetic fields, of excitons in MoS2, MoSe2, and WS2 monolayers. They follow the diamagnetic shift of the series of exciton Rydberg states to extract accurately and in a quite consistent manner the effective masses and binding energies of excitons (and some other material parameters). The authors have performed an excellent experimental analysis, with a massive amount of careful measurements. The data analysis is sound. It relies on the well understood behavior of excitons in a magnetic field and on the broadly adopted model of the confined potential (Rytova-Keldysh model) to model the spectrum of excitonic excited states.

The key results are summarized in Table 1, where one could find the most fundamental parameters of excitons in four archetypical direct-gap monolayer semiconductors - MoS₂, MoSe₂, WS₂, and WSe₂ (for this material, the authors cite the results from their previously reported study, Ref. [27]). These materials are the foundational blocks for the rapidly expanding field of TMDC-based heterostructures. Thus, the reported results are very timely and will certainly be of interest for the broad community of researchers working on TMDCs. However, I have several questions and remarks, which I ask the authors to address before I recommend the results to be published in Nature Communications:

Reply: We appreciate the reviewer's detailed reading and very positive assessment of the manuscript, and in particular for noting that "...the results are very timely and will certainly be of interest for the broad community of researchers working on TMDCs." Moreover, his/her constructive feedback is most welcome, and as detailed below we answer/accommodate all questions and suggestions.

(1) This study reports the band gap for WS₂ (MoS₂) to be about 2.238 eV (2.160 eV) with the accuracy of +/- 3 meV. In 2016, the authors published in Nature Communications a similar study of WS₂ and MoS₂ monolayers on Si/SiO₂. For WS₂, they reported 410 meV binding energy and ~ 2.044eV exciton energy. This would give the bandgap at about 2.45eV, i.e. 200meV higher compared to the value reported here. Similar discrepancy can be found for MoS₂. Of course, the TMDC monolayers on hBN and on SiO₂ are exposed to different dielectric environment, so the binding energy could be different. However, this should not affect the value of the single-particle band gap. Since the authors employed similar experimental methods and analyzed the data within the same Rytova-Keldysh model, I would expect to see in this manuscript a discussion comparing these two studies.

Reply: In fact, the dielectric environment surrounding a TMD monolayer does have quite a significant influence on both the exciton binding energy *and* the single-particle band gap. To leading order, both decrease by about the same amount as dielectric screening increases (which is why the exciton's transition energy is observed to change very little). Many papers have calculated the effects of the bandgap renormalization due to dielectric screening; see for example [Winther&Thygesen, 2D Materials **4**, 025059 (2017)], or also Refs. [44], [49], [50], [51] in original manuscript. The very different dielectric environment of the monolayers in the present manuscript (full hBN encapsulation) is the reason that the reported single-particle gaps are smaller as compared to previous work. We have added a sentence to the revised manuscript to make this point more clear.

(2) On pages 7 (last paragraph) and 8, the manuscript shows and discusses two unexpected properties of the 1s exciton in monolayer MoS₂ - asymmetric broadening at high field and the small valley Zeeman g-factor. The authors suggested that the high-energy shoulder lying at ~15meV above the main peak could be the same feature observed in zero-field spectra in Ref.24 and its origins could be related to dark/grey excitons. I don't think the data presented in this manuscript or in Ref.24 allow to draw such conclusions. Dark excitons should have the g-factor two times or more larger than the bright one. If the dark exciton is at 15meV above the bright exciton at zero field, then it should shift away rapidly and won't be at the shoulder position observed in this work.

Reply: Actually, the manuscript does not draw any such conclusions. Rather, the manuscript clearly states that the origin of the observed high-energy shoulder is not yet known, and also that its relationship to 'dark' states is only a suggested possibility ("*Although its origin is not yet known, one possibility is that this high-energy shoulder may be related to the presence of ... dark or grey excitons states*"). Nowhere does the manuscript insist or conclude that this must be the case. As is normal when presenting new data, we feel that it is reasonable to suggest potential explanations for phenomena that are not fully understood. Nonetheless, in the revised manuscript we downplay even further this potential scenario.

The experimental facts are that i) out of all the TMD semiconductors *only* MoS2 shows an anomalous g-factor and anomalous broadening of the 1s exciton, and ii) *only* MoS2 also shows an additional absorption feature near the 1s exciton (15meV above). It is therefore logical to point out that the anomalous behavior may be related to this nearby absorption. Although the origin of this nearby absorption is not known (as stated), and a coupling mechanism not clear, it *is* true that recent calculations [Ref. 69] indicate that dark excitons in MoS2 may lie slightly above the 1s bright exciton. Therefore we believe it is quite reasonable to suggest this as a potential possibility (especially in the absence of other obvious or plausible explanations). Finally, we note that dark exciton states are spin/valley degenerate at zero field, and at least one branch will shift towards (not away from) the bright exciton in applied fields.

(3) As for the surprisingly small g-factor of the 1s state in monolayer MoS2, the authors "tentatively suggest" that it may arise from interaction with dark excitons. Even considered as a possible mechanism, it requires some justification. Could authors discuss what/how the interaction with dark excitons would result in changing and specifically, in lowering the g-factor, and why this eventual interaction is absent or not effective in the lower quality samples.

Reply: Again, the manuscript only suggests that the observed small g-factor of the 1s state might be related to its proximity to the higher-lying states, and does not draw conclusions or insist on the (as-yet-unknown) origin of these states. Clearly, any form of interaction/mixing and subsequent level repulsion between the bright exciton σ - branch and these higher-lying resonance would result in a smaller measured g-factor (because the σ - state would be pushed “away” from the higher resonance). Essentially this is what the manuscript already says (“*we tentatively suggest that the small splitting...may arise from close proximity to and interaction with nearby states...*”). Nonetheless we agree that any relationship between these nearby states and dark excitons is speculative and in the revised version we further downplay the suggested role of dark excitons. We note further that in general mixing between two states can arise from Rashba-type mechanism which could depend on sample quality. Of course this scenario is also rather speculative and requires more investigation.

(4) The color map in Fig. 4 seems to show that the 3s line is stronger than the 2s one (on the +65T side, in particular). This is totally unexpected if this is the case. Could the authors check the intensities of 2s and 3s lines at +65T and, of course, if the unusual ratio of 2s/3s intensities is confirmed, discuss its possible origins.

Reply: The A:3s absorption feature is weaker than that of the A:2s. It may appear otherwise in the color map of Fig. 4b but that is only because A:3s is superimposed on the already-strong B:1s absorption. It is possible to see from the inset of Fig. 4a that while A:2s is quite clear at zero field, A:3s is barely discernable. Moreover, note that at high fields a comparison is precluded because A:2s merges with B:1s. For fields ~ 30 T where both peaks are discernable, A:3s is indeed less strong, in line with expectation.

(5) A very important starting point of this study is the claim that the measurements were performed on charge neutral samples. This very well could be the case. However, a more correct statement would be that the samples were not intentionally doped. What about the unintentional doping? As far I know, it is always necessary to work with gated samples and apply the gate voltage to bring the sample to the charge neutrality point. While the high-field pulsed experiments are hardly to be performed on gated samples because of the extreme nature of the pulsed field environment, it is important to estimate the carrier density in the studied (or similar) samples. Moreover, this analysis is necessary to support the discussion about the origins of "heavier-than-expected" excitons in Mo-based monolayers, which may be an intrinsic property of their bandstructure (as proposed here) or resulted from e-e interactions at (somewhat) higher carrier density. I am wondering if the authors performed or could perform PL characterization. The presence (or absence) of charged excitons may indicate that the samples are doped (or intrinsic).

Reply: The absorption spectra themselves indicate (effectively) zero doping in the monolayers: Namely, the spectra show only a single absorption peak from the neutral “A” exciton, and no indication of any additional lower-energy absorption from either negatively or positively charged exciton peaks. Similar to PL, any significant level of doping (intentional or unintentional) would give rise to lower-energy absorption features due to charged excitons – and significant quenching of the neutral exciton absorption -- as is well established from many previous studies of gated hBN-encapsulated monolayers [see for example Ref. [43] of original manuscript, or Fig. 3 of *PRB* **99**, 085301 (2019), or Supp. Fig S4 of *Sci. Adv.* **5**, eaau4899 (2019)]. Our spectra show only neutral exciton absorption. In addition, low-temperature PL spectroscopy of nominally equivalent hBN-encapsulated TMD monolayers (prepared in the same lab by the same methods in Toulouse) confirm that the PL originates almost entirely from the neutral exciton and not charged excitons, again indicating charge-neutrality. We have added a sentence to the revised manuscript to emphasize this point.

(6) Overall, I found that the manuscript is long and overloaded with somewhat technical and/or redundant information. I think moving such details to Suppl. Materials, focusing more on the physics and shortening the main text could make the manuscript better and help it meet with the high standards of Nature journals. For example, it might be enough to say in the introduction that higher order exciton states have not been previously observed in specific materials and not to repeat it in the following sessions. It may not be necessary to show the results obtained on multiple samples in Figs.2 and 3, etc.. Also, the manuscript sounds to me employing repeatedly unnecessarily subjective and dramatic vocabulary (crucially, importantly, interestingly, ...).

Reply: We thank the reviewer for feedback on how to improve the manuscript. Following these suggestions, we have tightened up the focus of the manuscript by avoiding redundant information and moving the last section and the original Figure 5 (about the B:2s exciton in MoSe₂, which is somewhat tangential to the main point of the manuscript) to the Supplementary Information. We do keep, however, figures showing data from multiple samples (eg Figure 2), since this shows the robust reproducibility of the phenomena --which we feel is important-- and supports the validity of the general conclusions. Moreover, its presence does not significantly lengthen the manuscript text. Finally we note the inclusion of very new data on diamagnetic shifts of the ‘fifth’ monolayer TMD semiconductor, MoTe₂.

(7) The reference list seems to be detailed and up to date. Still, the authors may consider to add in the appropriate context the following references: Chen et al, Nano Lett. 19, 42464 (2019) - Rydberg excitons and Zeeman splitting in WSe₂; Koperski et al, 2D Mater. 6 015001 (2019).

Reply: The reference list is updated.

Reply to Reviewer #3:

The manuscript by M. Goryca et al. provides to the community a set of material parameters such a exciton reduced mass, exciton bohr radius, g-factors for three different two-dimensional semiconductors, namely WS₂, MoS₂ and MoSe₂. These informations are extracted from delicate low temperature, polarization resolved, absorption experiments in high magnetic fields (up to 91T) provided by a pulsed resistive solenoid. These results are obtained on high quality 2D semiconducting materials encapsulated in hexagonal boron nitride to strongly enhance the optical quality and reach the homogenous limit resulting in thin linewidths of the excitonic absorptions. The authors use a technique that they (part of the authors) have developped and used to study monolayers of WSe₂ (reported in Ref 27) : they build the van der Waals hetrostructure directly on the surface of the optical fiber and collect the transmitted light. In this manuscript, the authors successfully extend this technique to the other members of the 2D semiconductor

family. The main results appear to be: i) the experimental determination of reduced mass for these three compounds, and the discrepancy between the measured reduced excitonic masses and the ones predicted by DFT, for Mo based compounds. The introduction and the body of the manuscript include references to a large and relevant part of the literature. The data are of high quality, are well presented and commented.

Reply: We appreciate the Reviewer's nice summary of the manuscript and are very glad to read that he/she finds it of "high quality, well presented and commented". We emphasize that not only are the exciton masses measured for the first time in MoS₂, WS₂, and MoSe₂ (as the reviewer notes, and now also MoTe₂), but also we measure their relevant dielectric screening properties [Table 1]. These are equally-important inputs for the rational design of van der Waals heterostructures using TMD monolayers, and surely to be found of keen interest to the broad community of researchers working in the burgeoning field of 2D materials.

The case of MoS₂ is of particular interest and this manuscript adds some new data concerning this topic but doesn't provide any explanation (anomalously small g-factor for the 1s state, heavier masses than expected). Even if the B exciton spectrum overlaps with A exciton excited states due to the smaller valence band splitting (also in MoSe₂), exciton excited states can be observed, allowing for the analysis of the excited state sequence through the RK potential. No special originality here compared to the study reported on WSe₂ by different groups, including some of the authors.

Reply: We agree that the behavior of MoS₂, as newly revealed by these high-field measurements, is particularly interesting. Actually the family of Mo-based TMDs, which have larger masses and overlapping A:ns and B:1s excitons, significantly benefits from the use of even larger magnetic fields as compared to past work on W-based monolayers. That is why we needed the 90+T magnetic fields to resolve the shifts in MoS₂. We do believe that use of such unprecedented field range to study the monolayer TMD semiconductors does represent a certain degree of originality and novelty... Also, we note that the paper does discuss the possible origin of the anomalous g-factor of the 1s exciton in MoS₂; namely its proximity to and interaction with the higher-lying absorption features (15meV above).

The sample used are encapsulated in hBN which gives in PL emission linewidth in the 1-2 meV (FWHM). Here, the authors perform absorption experiments on similar structure and show absorptions for the A:1s closer to 10 meV in FWHM. Could the authors comment on this point? In general, could the author indicate the FWHM that they measure in absorption for A:1s, A:2s etc ... (this is complicated to do on the figures provided by the authors because of the scale) As I understand, the technique to produce the samples doesn't allow to move the sample on the fiber after transfer which makes it difficult to select the best, "bubble free", locations.

Reply: Yes – the revised manuscript now includes the measured FWHM linewidths of the main absorption peaks of our monolayer-on-fiber samples, which are as narrow as ~6 meV (for MoSe₂) and ~10 meV (for MoS₂ and WS₂). The Reviewer is also correct that our monolayer-on-fiber approach (necessary for pulsed field measurements) does not allow the freedom to choose the best spot on the structure, and this is the reason why the linewidths, while quite good, are not as narrow as the best reported results. Very narrow (~2 meV) PL lines have been recently reported by some groups for hBN-encapsulated TMD monolayers. However these PL data are typically acquired by confocal microscopy, which allows the experimental freedom to search the entire sample and find the spot with best optical quality (as the Reviewer probably knows, even hBN-encapsulated samples show significant spatial inhomogeneity). Moreover, confocal PL measures a very small spot (~1 micron diameter), which also reduces inhomogeneous spatial broadening. In contrast, single-mode optical fibers have ~3.5 micron diameter cores, and so the area of sample that we probe is ~10x larger, and therefore subject to increased spatial inhomogeneity. (We note that nominally identical samples assembled by the same methods in the same lab at Toulouse do exhibit very narrow (1-2 meV) linewidths -- in selected locations -- when probed by confocal microscopy.)

The authors also describe the field evolution of the B:1s and 2s state in MoSe2, observable in their high quality sample. They indicate a diamagnetic shift of $1.3 \mu\text{eV}/\text{T}^2$, significantly larger than for the A:1s. There is a problem here, probably a typo, because the etxt referes to the B:2s, Fig 5b shows the evolution of a peak around 2.017 eV but it is labeled B:1s. Let's assume that this is a typo and that Fig 5b refers to B:2s, then what is the diamagnetic shift of B:1s ? how does it compare to the one of A:1s ?

Reply: We are grateful to the Reviewer for catching this typo – indeed, the label on Fig. 5b should be “B:2s”, not “B:1s”. To answer the reviewer’s question, unfortunately the B:1s peak is too broad and weak (see Fig. 4a) and its diamagnetic shift is too small to measure accurately. Such a measurement of the B:1s diamagnetic shift is further hindered by its overlap with the A:3s absorption line and also by the proximity to the A:2s state, which makes accurate fitting unreliable. Therefore we can quantitatively and accurately compare only the diamagnetic shifts of B:2s state and the A:2s state. We have added an explanation of this point to the revised manuscript. (but also please note that the discussion of the B:2s exciton is being moved to the Supp. Material per feedback from Reviewer #2).

For me, this is a very nice experimental work bringing to the community a set of parameters describing the electronic and optical properties of the most studied 2D semiconducting materials, but appears largely as a follow up of Ref 27, extended to other materials. I cannot find a clear and original message delivered by this paper and, to my opinion, it should be published in a more specialized journal.

Reply: As stated in the Abstract, Introduction, and the Conclusion of the manuscript, the main goal and primary new discovery of this work is to provide the *first comprehensive and quantitative picture* of the fundamental exciton and dielectric properties of the most widely-studied Mo- and W-based monolayer semiconductors – *these are essential optoelectronic material properties that have never been measured experimentally until now*. Given the huge and worldwide interest in these monolayer semiconductors amongst physicists, chemists, and materials scientists -- not to mention tremendous parallel interests in assembling van der Waals heterostructures of these monolayers – we strongly believe that the first determination of the fundamental mass and dielectric parameters will indeed be very well and very widely received by an extremely broad community of scientists. *These parameters are required inputs for any rational design of optoelectronic devices incorporating monolayer TMDs*. Thus we maintain our viewpoint that Nature Communications represents the ideal venue for these new results. These results go well beyond what was reported in [27], which concerned only WSe2 and did not explicitly measure dielectric screening properties. Moreover the additional new findings of unexpectedly heavy electron masses in *all* Mo-based monolayers (even at charge neutrality, unlike transport studies), and the anomalously small g-factor in MoS2 also represent new and original results, which should certainly capture the attention of the many theoreticians working on density-functional theory (DFT) of these materials, and motivate improved models. And finally note that the inclusion of the new MoTe2 data provides a complete survey of the entire family of Mo-based monolayer semiconductors, from which general trends can now be experimentally confirmed and established.

* * *

List of changes to the manuscript

- Added very new high-field data on the final Mo-based monolayer TMD semiconductor, MoTe₂. This now appears as the last figure (+ discussion) in the revised manuscript. Together with the existing data on MoS₂ and MoSe₂, this completes the entire family of Mo-based monolayer TMDs. General trends are now established (eg, the increase of exciton mass and dielectric screening as the chalcogen atom becomes heavier, and the persistence of unexpectedly heavy masses in all Mo-based TMD monolayers).
- Added a comparison of our results to available experimental ARPES measurements of hole masses in MoS₂ and MoSe₂. This further highlights and supports the interpretation of anomalously large electron masses indicated by our results.
- Added a brief discussion of anomalously heavy electron and exciton masses, in context of recent theories of electron-phonon coupling in these monolayer semiconductors.
- Moved original Figure 5 and associated discussion of B:2s exciton to Supplemental Information (per Reviewer #2 feedback). Though still novel, discussion of the “B” excitons is somewhat tangential to the main focus of the paper. Further it is explained that we can’t measure the diamagnetic shift of B:1s.
- Per feedback from Reviewer #2, we downplay and further minimize the potential/suggested role that might be played by dark excitons in MoS₂.
- Per Reviewer #3, the revised manuscript now includes the FWHM linewidths of the exciton absorption peaks.
- The reference list is updated (per Reviewer #2).
- Added sentence stating that quasi-zero doping is verified by absence of trion absorption (per Rev. #2)
- Added sentence stating that we compare our data to state-of-the-art theory, eg ref [10] by Kormanyos (per Reviewer #1)
- Added phrasing that further emphasizes that dielectric screening changes both binding energies *and* bandgap (per Reviewer #2).
- Throughout the paper, we removed redundant information and moderated subjective vocabulary (per Reviewer #2).

REVIEWERS' COMMENTS:

Reviewer #1 (Remarks to the Author):

After going through the revised version of the manuscript, I find that the paper has significantly improved (MoTe₂ has been added on which literature is scarcely available). I also feel that the authors have answered satisfactorily to my criticism. The paper can be accepted in Nat. Commun. in the present form.

Reviewer #2 (Remarks to the Author):

The authors have answered my questions and have revised the manuscript properly. Also, I think the authors have well responded to other referees' comments and questions. I recommend the publication.

Reviewer #3 (Remarks to the Author):

I read the new version of the manuscript by Goryca et al. together with the response of the authors to the different queries/comments of the referees. Adding data and discussions on MoTe₂ monolayers and their excitons really improves the manuscript and indeed provides a complete set of comparable high quality data on the most representative members of the family of semiconducting TMDC. There is now a clear picture showing that Mo based TMDC monolayers are still to be understood at a deeper level to allow for accurate calculations of their basic electronic properties (mass, gaps, binding energies, ...)

Also the authors have included in the new version of their manuscript all the points raised by the different referee which I think increases the quality of the discussion. These results are obtained/reproduced on different samples, are in line with previous results obtained on WSe₂ and already published by different groups, which for me rules out any doubts on their validity.

I think the new version of the manuscript is suitable for publication in Nature Communications as it is. I am now convinced that it will become an extremely useful reference in the field of TMDC.